



# Evaluation of the first year of Pandora NO$_2$ measurements over Beijing and application to satellite validation

Ouyang Liu[1, 2], Zhengqiang Li[1,*], Yangyan Lin[2,3], Cheng Fan[1], Ying Zhang[1], Kaitao Li[1], Peng Zhang[4], Yuanyuan Wei[1], Tianzeng Chen[2,4], Jiantao Dong[1,5], Gerrit de Leeuw[1,6]

[1] State Environmental Protection Key Laboratory of Satellite Remote Sensing & State Key Laboratory of Remote Sensing Science, Aerospace Information Research Institute, Chinese Academy of Sciences, Beijing 100101, China
[2] University of Chinese Academy of Sciences, Beijing 100049, China
[3] Key Laboratory of Watershed Geographic Sciences, Nanjing Institute of Geography and Limnology, Chinese Academy of Sciences, Nanjing 210008, China
[4] State Key Joint Laboratory of Environment Simulation and Pollution Control, Research Center for Eco-Environmental Sciences, Chinese Academy of Sciences, Beijing 100085, China
[5] Satellite Application Center for Ecology and Environment, Ministry of Ecology and Environment of People's Republic of China, Beijing 100094, China
[6] Royal Netherlands Meteorological Institute (KNMI), R & D Satellite Observations, 3730 AE De Bilt, The Netherlands

*Correspondence to*: Zhengqiang Li (lizq@radi.ac.cn)

**Abstract.** Nitrogen dioxide (NO$_2$) is a highly photochemically reactive gas, has a lifetime of only a few hours, and at high concentrations it is harmful to human beings. Therefore, it is important to monitor NO$_2$ with high-precision, time-resolved instruments. To this end, a Pandora spectrometer has been installed on the roof of the laboratory building of the Aerospace Information Research Institute of the Chinese Academy of Sciences in the Olympic Park, Beijing, China. The concentrations of trace gases (including NO$_2$, HCHO, O$_3$) measured with Pandora are made available through the open-access Pandora data base (https://data.pandonia-global-network.org/Beijing-RADI/Pandora171s1/). In this paper, an overview is presented of the Pandora NO$_2$ data collected during the first year of operation, i.e., from August, 2021, to July, 2022. The data show that NO$_2$ concentrations were high in the winter and low in the summer, with diurnal cycle where the concentrations reach a minimum during day time. The concentrations were significantly lower during the 2022 Winter Olympics in Beijing, showing the effectiveness of the emission control measures during that period. The Pandora observations show that during northerly winds clean air is transported to Beijing with low NO$_2$ concentrations, whereas during southerly winds pollution from surrounding areas is transported to Beijing and NO$_2$ concentrations are high. The contribution of tropospheric NO$_2$ to the total NO$_2$ VCD varies significantly on daily to seasonal time scales, i.e., close to 50% in autumn and winter, and close to 70% in spring and autumn. The comparison of Pandora-measured surface concentrations with collocated in situ measurements using a Thermo Scientific 42i-TL Analyzer shows that the Pandora data are low and that the relationship between Pandora-derived surface concentrations and in situ measurements are different for low and high NO$_2$ concentrations. Explanations for these differences are offered in terms of measurement techniques and physical (transport) phenomena. The use of Pandora total and tropospheric NO$_2$ vertical column densities (VCDs) for validation of collocated TROPOMI data, resampled to 100×100 m$^2$, shows that although on average the TROPOMI VCDs are slightly lower, they are well within the



35 expected error for TROPOMI of $0.5 \ \mathrm{Pmolec} \cdot \mathrm{cm}^{-2} + (0.2 \ \text{to} \ 0.5) \cdot VCD_{trop}$ The location of the Pandora instrument within a sub-orbital TROPOMI pixel of $3.5 \times 5.5 \ \mathrm{km}^2$ may result in an error in the TROPOMI-derived tropospheric $NO_2$ VCD between 0.223 and 0.282 $\mathrm{Pmolec.cm}^{-2}$, i.e., between 1.7% and 2%. In addition, the data also show that the Pandora observations at the Beijing-RADI site are representative for an area with a radius of 10 km.

## 1 Introduction

40 $NO_2$ is a trace gas that plays an important role in atmospheric chemistry (Seinfeld and Pandis, 1998), such as the $O_3$-$NO_x$-VOC sensitivity (e.g., Wang et al., 2021; Liu and Shi, 2021), the formation of aerosols (Behera and Sharma, 2011) and thus air quality (de Leeuw et al., 2021). $NO_2$ is also a precursor for the production of aerosol and ozone and therefore indicated as an essential climate variable (ECV) (https://gcos.wmo.int/en/essential-climate-variables/precursors; last visited 11th July 2023). Sources of tropospheric $NO_2$ include anthropogenic sources such as coal-fired power plants, motor vehicle emissions

45 and industrial chemical production (Carslaw and Beevers, 2005; Felix and Elliott, 2014), and natural sources such as stratospheric $NO_x$ intrusion, bacterial and volcanic action (Mather et al., 2004), and lightning (Zhang et al., 2020). $NO_2$ is removed by chemical reactions and diluted by horizontal and vertical transport (Jorba et al., 2012; Dai et al., 2022). Thus, the $NO_2$ concentration decreases with distance to the emission sources. Most of the $NO_2$ is located at altitudes between 0 and 3 km with a small amount in the upper troposphere and stratosphere due to chemical reactions such as the oxidation of nitrous

50 oxide (Brasseur and Nicolet, 1973; Grenfell et al., 2006; Dirksen et al., 2011; Herman et al., 2019). $NO_2$ is a short-lived trace gas with a chemical lifetime of $3.8 \pm 1.0$ h during the summer (Liu et al., 2016), increasing during colder conditions in winter and at higher altitudes (Herman et al., 2018). Together with the multitude of localized (power plants, factories) and diffuse (traffic) sources, this short lifetime results in a large spatial and temporal variation of the $NO_2$ concentrations (Weber and Bylicki, 1987; Sivakumaran et al., 2001; Zhao et al., 2020). To catch this variability, observations are needed with high

55 temporal resolution together with good spatial resolution.

Information on $NO_2$ concentrations can be obtained by using different methods and techniques, i.e., ground-based in situ or remote sensing measurements or satellite remote sensing. Each of these techniques provides different types of data with their own advantages and disadvantages. Instruments used for ground-based observations have the advantage that they can easily be serviced, have high and well-known accuracy, can measure continuously and thus provide good temporal coverage during

60 day and night. In China, ground-based in situ measurements of $NO_2$ concentrations are available, together with other species, from the National Real-time Air Quality Publishing Platform public website for air quality (AQ) monitoring data, maintained by the China National Environmental Monitoring Center (CNEMC) of the Ministry of Ecology and Environment of China (http://www.cnemc.cn/; last access 11th July 2023) (Zhai et al., 2019; Li et al., 2020; Xie et al., 2005). This network includes more than 2000 stations all over China, but they are mainly located in densely populated and urban centers. In situ data are

65 representative for concentrations near the surface and within a certain distance from the observation site, especially for a short-lived species like $NO_2$ with many local emission sources such as traffic using fossil fuel powered engines and



households. This leaves large gaps in the data coverage, i.e., there are no data for large areas outside the urban and industrial agglomerations.

Ground-based remote sensing of atmospheric $NO_2$ concentrations can be made using instruments such as MAX-DOAS
(Wagner et al, 2010) or Pandora (Herman et al., 2009). Both instruments are spectrometers with a mode viewing the sun directly, and another one viewing scattered radiation (sky) at different angles, providing information on the vertical distribution. In the current study, data from a Pandora instrument will be used (see Section 2.2 for more detail on the instrument).

The gaps in the spatial distributions can be filled with data from satellite observations, i.e., using dedicated satellite-based
instruments (spectrometers) providing wide spatial coverage. Such instruments have been developed during the last three decades with enormous improvement in both spatial resolution and accuracy, i.e., from the Global Ozone Monitoring Experiment (GOME, launched on the European Remote Sensing satellite ERS-2 in 1995), the SCanning Imaging Absorption spectroMeter for Atmospheric CartograpHY (SCIAMACHY; launched on the Environmental Satellite ENVISAT in 2002), the Ozone Monitoring Instrument (OMI; launched on the AURA satellite in 2004), GOME-2 (first launched on the
Meteorological Operational satellite METOP-A in 2006), to the TROPOspheric Monitoring Instrument (TROPOMI; launched on the Sentinel-5 Precursor (S-5p) satellite in 2017). However, although these sensors provide data with daily near-global coverage, they have only one single overpass each day (more at high latitudes) which does not capture diurnal variation (Bovensmann et al., 1999; Burrows et al., 1999; Levelt et al., 2006; Levelt et al., 2018; Verhoelst et al., 2021). The primary $NO_2$ product from such satellites is the total vertical column density, (VCD) (i.e., column-integrated $NO_2$
concentrations) which can be separated into tropospheric and stratospheric VCDs by using a chemical transport model (e.g., van Geffen et al., 2021). The accuracy of satellite data is usually less good than from ground-based data because corrections need to be made for contributions from the surface and from other atmospheric constituents contributing to the reflected solar radiation measured by the instrument at the top of the atmosphere (TOA). In addition, satellite data need regular validation to account for post-launch degradation. In this study, TROPOMI data will be used as discussed in Section 2.2.

For the validation of satellite-retrieved $NO_2$ data, the PANDONIA Global Network (PGN) has been established in 2018 to provide "long-term quality observations of total column and vertically resolved concentrations of a range of trace gases" (http://www.pandonia-global-network.org/, last accessed: 11th July 2023). The information on $NO_2$, $O_3$ and HCHO provided by PGN is obtained using Pandora instruments and is publicly available. Most PGN sites have been established in North America and Europe, with fewer data from Japan, South Korea and some other countries, thus leaving large areas uncovered.
In particular, until recently there were no publicly available data in China.

As a first step to fill this gap, a Pandora instrument has been installed in Beijing on the roof of the Aerospace Information Research Institute of the Chinese Academy of Sciences (AirCAS) laboratory building in the Olympic Park of the Chinese Academy of Sciences, Beijing, China (116.3786E, 40.0048N). The instrument has been running continuously since the end of July 2021. The observations and data processing follow the Pandora protocol (Cede et al., 2021) and provides high-
100 precision trace gas vertical column density (VCD) data in near real-time. The Pandora instrument, referred to as Beijing-





RADI, is the first instrument in China which joined the PGN network and all data are publicly available via the PGN website (https://data.pandonia-global-network.org/Beijing-RADI/Pandora171s1/, last accessed: 11th July 2023) within one day of the observations.

In this paper we provide an overview of the results from the first year of NO₂ observations from the Beijing-RADI Pandora instrument, i.e., total VCDs, tropospheric VCDs and surface concentrations. The site and experimental methods are discussed in Section 2. The results are presented in Section 3, where we discuss the variations of NO₂ VCD and surface concentrations and the contribution of tropospheric NO₂ to the total VCD on hourly, daily, monthly and seasonal time scales. Influences of wind speed and direction on the NO₂ concentrations at the Beijing-RADI site are discussed. Pandora-derived surface concentrations are evaluated by comparison with collocated in situ observations and differences are explained in terms of local production and long-range transport. Pandora NO₂ tropospheric and total VCD are used for validation of TROPOMI NO2 data and the spatial representativeness of the Pandora is evaluated. A brief summary of the study and the main conclusions are presented in Section. 4.

## 2 Data and methodology

### 2.1 Site description

Beijing is a Megacity with a population of over 21 million, estimated to grow to 25 million in 2023 (https://worldpopulationreview.com/world-cities/beijing-population, last accessed: 11th July 2023). The vast majority of the people in Beijing live within the fifth ring road (see Figure 1). Beijing is located in the north of the North China Plain (NCP), with the Yanshan and Taihang Mountains to the north and west, respectively. The NCP includes a highly industrialized and urbanized area where the weather conditions are often conducive to the accumulation of air pollution resulting in the frequent occurrence of haze episodes (e.g., Sundström et al., 2012; Sun et al., 2014; Li et al., 2016). The effects of pollution in the NCP on the air quality in Beijing varies with weather conditions. Southern airflow transports pollution to the city, whereas northern airflow transports clean air from the north. Hence the air quality situation may vary strongly, depending on large scale meteorological patterns (e.g., Hou et al., 2020). Furthermore, as the capitol city of China, Beijing hosts the Chinese government and many national and international events. During such events, emissions from industry and traffic in and around Beijing are regulated to improve air quality (e.g., Wang et al., 2010; Wang et al., 2016; Fan et al., 2021; Chu et al., 2022).

The AirCAS laboratory building is located at the north side of Beijing between the Fourth and Fifth Ring Roads (see Figure 1). The Beijing-RADI site is located at the roof of this building, at a height of 21m above the ground. The site includes instrumentation for atmospheric observations such as a sun photometer, a lidar and a suite of instruments for in situ measurements of aerosol properties, trace gas concentrations, and for solar radiation. The Pandora instrument has been added in 2021, to measure NO₂ VCDs and surface concentrations. Due to its location in a science park at the edge of the Olympic Park, the Beijing-RADI site is considered a combined urban and suburban background observatory with small local



emissions and large influences from nearby highways, urban activities and industrial pollution advected from the south of Beijing and the NCP.

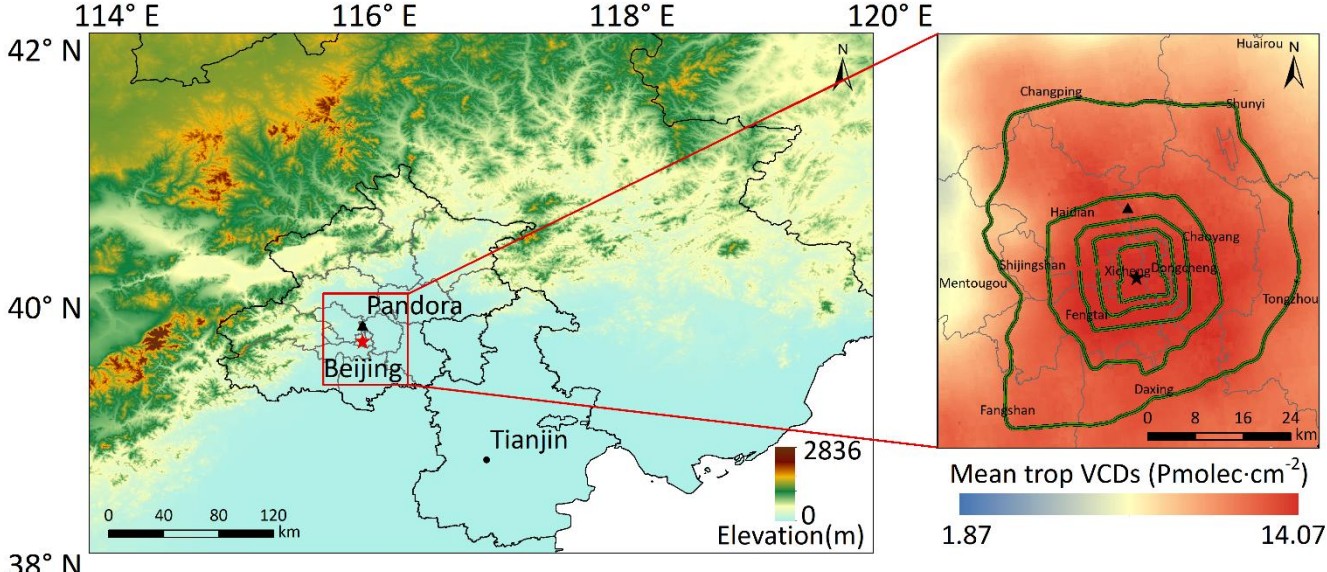

**Figure 1: Digital elevation map of Beijing and surroundings. The solid black line is the provincial boundary and the solid grey line is the administrative boundary of Beijing on a district basis. The Beijing-RADI site is indicated by the black triangle, at about 12 km north of the administrative center of China (red pentagram). The inset shows Beijing with the ring roads 2-6 (from the center to outside; there is no ring road 1) overlaid on the TROPOMI-derived NO₂ tropospheric VCD re-gridded to a resolution of 100×100 m² and averaged over the period from August 1, 2021, to July 31, 2022.**

## 2.2 Instrumentation and auxiliary data

### 2.2.1 Pandora

Pandora is a sun-viewing instrument at solar zenith angles (SZA) smaller than 80°. The instrument consists of a UV/VIS spectrometer connected to an optical head through a 400 µm core diameter fiber-optic cable (Herman et al., 2009; Cede, 2021). The optical head is mounted on a sun tracker for accurate pointing at the sun (precision 0.013°; Herman et al., 2009). Solar radiation is collected by the front-end optics with a field of view (FOV) of 2.6° for direct-sun observations using a diffuser and with a FOV of 1.5° for sky observations without a diffuser. The vast majority of the light sampled in direct observation mode comes from an angle of 0.5°. The received solar radiation is transmitted through a fiber-optic cable to the spectrometer which is a 2048×64 pixels back-thinned Hamamatsu charge-coupled device (CCD), with a 50 µm entrance slit, and a grating with 1200 lines/ mm.

Pandora measures spectrally resolved solar radiances at wavelengths between 290 nm and 380 nm, using a UV bandpass filter, and between 280 nm and 525 nm, with a spectral resolution of 0.6 nm and 4.5 times oversampling. In clear-sky conditions, about 4000 spectra are measured in about 80s, including about 20s of dark current measurements between each spectral measurement (Herman et al., 2009; Cede, 2021). These 4000 spectra are averaged to achieve very high signal to



noise ratios (Herman et al., 2019). The spectra are used to determine trace gas amounts using the differential optical absorption spectroscopy (DOAS) technique, i.e., spectral fitting, as described in detail in the ATBD (Cede, 2021). For the retrieval of $NO_2$ VCDs, the part of the spectrum between 400 and 440 nm is used.

Using direct sun measurements, information is obtained on the total VCDs of the trace gases. Diffuse (scattered) radiation is measured at 5 pointing zenith angles (PZAs) in sky mode which, together with the direct sun measurement, provides

information on the tropospheric VCD and on the surface concentrations. The PZAs are 0º, 60º, 75º, 88º and a maximum angle taken as 89º. The measurements are taken in a V shape (all angles are measured twice around a central angle) as described in Cede (2021).

Details of the Pandora spectrometer instrument can be found from the Pandora project website https://Pandora.gsfc.nasa.gov/Instrument/ (last accessed: 11th July 2023), as well as the NASA Pandora website

https://avdc.gsfc.nasa.gov/pub/DSCOVR/Pandora/Web_Pandora/index.html (last accessed: 11th July 2023). The Beijing-RADI Pandora is operated following the PGN operational procedures and data processing, described in detail in Cede (2021), which also includes the ATBD. In this study, L2 data products for $NO_2$ (L2H files L2Tot for total VCD and L2Tro for tropospheric NO2 and surface concentrations) are used which include all measurements available. The data quality (DQ) has been checked for a number of criteria which have to be satisfied. Only DQ0 (assured high quality), DQ1 (assured medium

quality), DQ10 (not-assured high quality) and DQ 11 (not-assured medium quality) data are used. DQ2 and DQ12 data (low quality) are available in the data files but their use is not recommended (Cede, 2021). The estimated clear-sky precision of the $NO_2$ total VCD product is 0.269 $Pmolec \cdot cm^{-2}$ with a precision of about $\pm 1.3$ $Pmolec \cdot cm^{-2}$ (1 $Pmolec \cdot cm^{-2} = 1 \times 10^{15}$ $molec \cdot cm^{-2} = 3.745 \times 10^{-2}$ DU) (Herman et al., 2009).

### 2.2.2 TROPOMI

TROPOMI on-board the Sentinel-5 Precursor (S-5p) satellite of the European Space Agency (ESA) was launched on 13 October 2017 (Veefkind et al., 2012) to fill the gap between OMI (Schoeberl et al., 2006; Levelt et al., 2018) and future payloads. TROPOMI has been designed to retrieve the slant column densities (SCDs) of key atmospheric species such as $NO_2$、$O_3$、$SO_2$、 HCHO, $CH_4$ and CO (e.g., http://www.tropomi.eu/) with a spatial resolution of 3.5×7 $km^2$ which was further reduced to $3.5 \times 5.5$ $km^2$ from 6 August 2019. This spatial resolution is suitable for air quality monitoring at city

level (Lama et al., 2020). TROPOMI has a swath width of 2600 km which allows for daily global coverage, with an overpass time at about 13:00 local time (Van Geffen, 2021). Trace gas parameters are retrieved using six spectral channels covering wavelengths from the ultraviolet to the near-infrared. The $NO_2$ total SCD is retrieved by application of the DOAS method to the spectral radiation and irradiance measured in the UV-VIS (320-500nm) spectral channels (Van Geffen et al., 2015; Boersma et al., 2011). The stratospheric and tropospheric SCDs are derived from the total SCD by using the TM5-MP

chemical transport model (CTM) (Dentener et al., 2003). The meteorological information required to run the model is obtained from the European Centre for Medium-Range Weather Forecast (ECMWF) (Dee et al., 2011; Uppala et al., 2005).



The stratospheric and tropospheric SCDs are converted to VCDs using a height-dependent air mass factor (AMF) look-up table with a spatial resolution of 1° × 1° built using the double adding KNMI (DAK) radiative transfer model (Palmer et al., 2001; Boersma et al., 2004). It is worth noting that the $NO_2$ total VCDs provided by TROPOMI are the sum of tropospheric and stratospheric VCDs calculated as described above. The margin of error given by the data provider is 0.5 Pmolec · $cm^{-2}$ + (0.2 to 0.5) · $VCD_{trop}$ (Van Geffen, 2021).

TROPOMI provides near-real time data (NRTI) and off-line data (OFFL). The OFFL data used in the current study are produced using observational meteorological data for assimilation in the TM5-MP CTM (Van Geffen, 2021), whereas the NRTI are produced using meteorological forecast data. The data were produced using processor version 1.4.0. (van Geffen, 2021). Only data were used with high quality, i.e., with QA > 0.75, which disqualifies scenes with a cloud radiance fraction > 0.5, some scenes covered by snow or ice and scenes that have been determined to include errors or problematic retrievals. TROPOMI $NO_2$ VCD data were downloaded from the ESA website: https://s5phub.copernicus.eu/ (last accessed: 11th July 2023).

### 2.2.3 In situ measurements of $NO_2$ concentrations

In addition to $NO_2$ surface concentrations obtained from Pandora observations, data from an in situ instrument are available for a short period of time, i.e., from a Thermo Scientific 42i-TL Analyzer (https://www.thermofisher.com/order/catalog/product/42ITL, last accessed 11th July 2023). This instrument was located at the roof of the AirCAS building at approximately 20 m from the Pandora instrument, during the period January 10-30, 2022 (20 days). The Thermo Scientific 42i-TL Analyzer measures the chemiluminescence caused by the reaction of NO with ozone, where NO is produced from the dissociation of $NO_2$ ($2NO_2 \leftrightarrow 2NO + O_2$) on a heated molybdenum surface. The chemiluminescence intensity is proportional to the NO concentration (Kley and Mcfarland, 1980). More detailed information is provided in the instrument's manual (https://www.manualslib.com/manual/1251056/Thermo-Scientific-42i.html (last accessed: 11th July 2023). The Thermo Scientific $NO_2$ data used in this study were screened for high quality using the quality flags and anomalous data were discarded.

### 2.2.4 Re-analysis data

ERA5 hourly reanalysis wind data, available from the European Centre for Medium-Range Weather Forecasts (ECMWF, https://www.ecmwf.int/), were used in the data analysis. This dataset provides hourly atmospheric and oceanic information such as wind speed, temperature, and specific humidity at different pressure levels. ERA5 hourly wind data were downloaded from the ECMWF website (ERA5 web: https://cds.climate.copernicus.eu/cdsapp#!/dataset/reanalysis-era5-pressure-levels?tab=form, last accessed: 11th July 2023).





## 2.3 Methodology

For the evaluation of the NO₂ observations and the comparison between Pandora and TROPOMI NO₂ VCDs, the data need to be collocated. Furthermore, the difference in observation geometry needs to be taken into account. The Pandora bottom-up observations of direct and scattered (sky) solar radiation depend on the NO₂ vertical profile, the spatial distribution and the

SZA (Boersma et al., 2011; Van Geffen et al., 2015). For example, for a situation when most of the total VCD is located below 2 km and the SZA is 45º, the total viewing area of Pandora with a FOV of $(2.6\pi \, / \, 180) \cdot (2 \, / \cos{(SZA)})$, is about $128 \times 128$ m². The TROPOMI top-down observations from space provide NO₂ VCDs with a spatial resolution of 3.5×5.5 km². To match the observation volumes, the TROPOMI observations were resampled to a spatial resolution of 100×100 m², using Google Earth Engine (GEE, (https://developers.google.com/earth-engine/guides/scale; last accessed: 11th July 2023).

For an accurate comparison between TROPOMI and Pandora, only high quality data were retained for the collocated TROPOMI/Pandora pixels. For time collocation, Pandora data obtained within 10 minutes before and after the TROPOMI overpass time were averaged. The resulting collocated TROPOMI and Pandora data sets were quantitatively evaluated using scatterplots and the statistical metrics mean difference (MD), mean absolute difference (MAD), mean relative difference (MRD), standard deviation (σ), correlation coefficient) and the fitted slope.

MD is calculated by averaging the difference between two observations, as shown in Eq. (1):

$$MD = \frac{1}{n}\sum_{i=1}^{n}\left( VCD_{TROPOMI,i} - VCD_{Pan,i} \right) , \tag{1}$$

The MAD, given by Eq. (2), is defined as the mean of the absolute differences between TROPOMI and Pandora:

$$MAD = \frac{1}{n}\sum_{i=1}^{n}\left| VCD_{TROPOMI,i} - VCD_{Pan,i} \right| , \tag{2}$$

MRD, Eq. (3), is the mean of the differences between TROPOMI and Pandora when normalized with Pandora's VCDs.

Positive and negative values of the MRD indicate the degree of overestimation or underestimation:

$$MRD = \frac{1}{n}\sum_{i=1}^{n}\frac{VCD_{TROPOMI,i} - VCD_{Pan,i}}{VCD_{Pan,i}} \times 100\% , \tag{3}$$

The standard deviation σ is defined in Eq. (4).

$$\sigma = \sqrt{\frac{1}{n}\sum_{i=1}^{n}\left( VCD_{TROPOMI,i} - VCD_{Pan,i} \right)^2} , \tag{4}$$

The dilution factor, $D_f$, is defined in Eq. (5), where $VCD_{TROPOMI,0.1}(median)$ and $VCD_{TROPOMI,R}(median)$ represent the

median value of the TROPOMI VCD within a radius of 0.1 km and within a radius R around the location of Pandora, respectively (Chen et al., 2009; Griffin et al., 2019; Pinardi et al., 2020):

$$D_f = \frac{VCD_{TROPOMI,R}(median)}{VCD_{TROPOMI,0.1}(median)} , \tag{5}$$





## 3 Result and discussion

### 3.1 Pandora data overview

An overview of the Pandora data is presented in Figure 2, as time series for each month from August 2021 to July 2022, where the time scale is local time (i.e., Beijing Time or BJT, i.e., UTC+8) throughout this paper. Total $NO_2$ VCD, tropospheric $NO_2$ VCD and surface $NO_2$ concentrations are plotted in different colors and only high and medium quality data are included (DQ0, DQ1, DQ10 and DQ11). Time series including also low quality data (DQ2 and DQ12) are presented in the supplementary material, Figure S1. Although the use of low quality data is not recommended, we have added this figure

in the Supplementary material to show the high data density throughout the year. For the tropospheric VCDs, including low quality data adds 4681 data points to the total of 8620 data points during the one year study period (i.e., 54.3% of all data are low quality). Among the total VCDs, 21767 data points out of a total of 80153 (27.2%) are low quality. The comparison of Figures 2 and S1 shows that low quality data occur in particular during the winter months when the $NO_2$ concentrations are high and aerosol (PM2.5) concentrations are also high, the boundary layer is shallow, the sky is often cloudy, and air quality

is bad. These are conditions when the data processing flags bad situations for some criteria, the reliability of the results decreases and hence the DQ is increased to DQ2.

### 3.1.1 Diurnal, day-to-day and episodical variations

The data in Figure 2 show the variation of the $NO_2$ total and tropospheric VCDs and surface concentrations measured with Pandora during the first year of operation. Overall, the $NO_2$ concentrations are low in the summer and high in the winter, as

is commonly observed in satellite data (e.g., van der A et al, 2006; Wang et al., 2019; Fan et al., 2021). Monthly and seasonal variations will be further discussed in Section 3.1.2. The data show the obviously larger total VCDs than the tropospheric VCDs and the diurnal variation of all three parameters. Although Pandora has the capability of using moonlight for the $NO_2$ observations (Cede, 2021), the Beijing-RADI data includes only daytime (solar) measurements. The $NO_2$ VCDs decrease in the morning to reach a daily minimum around local noon and then increase. The complete diurnal cycle of

surface concentrations measured with the Thermo Scientific 42i-TL Analyzer, discussed in section 3.4, confirms these observations and shows that the maximum is reached during the night.

The time series in Figure 2 also show large variations of all three parameters during periods of several days, suggesting the occurrence of pollution episodes. An example is a long period from 11 to 19 November 2021, when the $NO_2$ parameters are increasing until 15 November, whereafter the total VCDs are not available and the tropospheric VCDs and surface

concentrations remain relatively high until 19 November. On other occasions the tropospheric VCDs and surface concentrations are not available, but the total VCDs increase over a period of several days. For example, this is observed in December 2021 during 5 different periods: 1-5 December, 7-9 December, 11-13 December, 16-22 December and 25-28 December. Comparison with data from the nearby national air quality monitoring site in the Olympic Forest Park in Beijing (Figure S2) shows that these periods coincide with similar behavior of the air quality (as indicated by the air quality index,





dominated by high aerosol (PM) concentrations) and the associated NO₂ time series (in situ). The variation of the NO₂ concentrations and the occurrence of haze episodes is strongly related to the wind direction and the wind speed, the influence of which will be discussed in Section 3.2.



**Figure 2: Overview of NO₂ data measured with Pandora at the Beijing-RADI site, as time series during each month from August 280 2021 to July 2022: total VCD in green and tropospheric VCD in red (both on left vertical axes in Pmolec.cm⁻²) and surface**



**concentrations in red (plotted on the right axis in µg.cm⁻³) Only data with DQ0, DQ1, DQ01 and DQ11 are plotted. Note that the surface concentrations scale has been chosen such that they are plotted at the bottom of the VCDs.**

### 3.1.2 Monthly mean NO₂ VCDs

As shown in Figure 2, apart from diurnal and short-term variations, there is also a clear seasonal variation of the NO₂ VCDs.
This is further illustrated in Figure 3 where time series are plotted for the mean and median of monthly averaged tropospheric VCDs derived from Pandora measurements during the period from August 2021 to July 2022. Only the data measured between 08:00 and 17:00 BJT, corresponding to daylight hours during the winter, have been included, with DQ0, 1, 10 and 11. This selection of high-quality data reduces the number of days for which data are available, but with ensured reliability. A similar figure including also DQ2 and DQ12 data is included in the Supplementary material (Figure S3) which
shows that in the autumn and winter the NO₂ VCDs are high and that they are low in the spring and summer, in good agreement with the seasonal variation which is commonly observed in satellite data (e.g., van der A et al, 2006; Wang et al., 2019; Fan et al., 2021). The number of days for which data are available for each month is included in Tables 1 (high quality only) and S1 (including DQ2 and 12). Comparison shows that mainly the monthly mean VCD for December 2021 is affected by the data selection: the number of days is reduced from 31 to 12 and apparently mainly data with high VCDs, i.e., during
days with bad AQ (Section 3.1.1), were removed by the DQ2 criteria. As a result, the annual cycle of the mean tropospheric NO₂ VCDs in not as obvious in the high-quality data as in Figure S3 which clearly shows the maximum occurring in the winter and the minimum in the summer. Such an annual variation pattern is in good agreement with near surface observations of in situ NO₂ measurements in Beijing. The mean, median, minimum and maximum values, and the standard deviation ($\sigma$), of the tropospheric NO₂ VCDs over Beijing for each month are summarized in Table 1. The difference
between the median and mean values is significantly larger in the winter months than in the summer. The variability of the NO₂ VCDs over Beijing is much higher during the winter than in the summer. The minimum in February 2022, as compared to January and March, illustrates the effectiveness of the emission control policy in and around Beijing during the 2022 Winter Olympics (Chu et al., 2022). Time series of the monthly mean total VCDs, are presented in Figure S4 and shows similar variation as for the tropospheric VCDs in Figure 3.

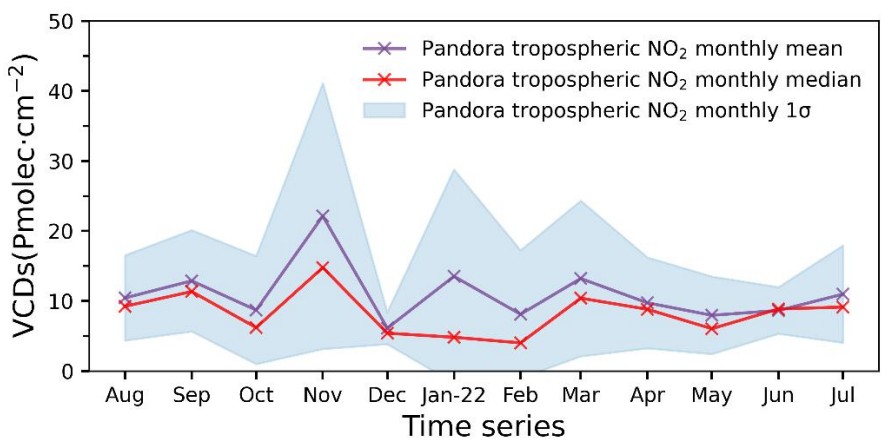






**Figure 3: Time series of the monthly mean and median tropospheric NO₂ VCDs over Beijing from August 2021 to July 2022. Only high quality data (DQ 0, 1, 10 and 11) are included, which limits the number of days for which data are available as indicated in Table S3. The blue shaded area indicates the standard deviation for the monthly mean data.**

**Table 1: Statistics of the Pandora-derived monthly mean NO₂ tropospheric VCDs. Only data with DQ0, 1, 10, 11 included.**

|  | Aug | Sep | Oct | Nov | Dec | Jan | Feb | Mar | Apr | May | Jun | Jul |
|---|---|---|---|---|---|---|---|---|---|---|---|---|
| Mean (Pmolec.cm⁻²) | 10.45 | 12.89 | 8.72 | 22.14 | 6.12 | 13.53 | 8.12 | 13.22 | 9.75 | 7.97 | 8.66 | 10.99 |
| Median (Pmolec.cm⁻²) | 9.26 | 11.36 | 6.22 | 14.78 | 5.42 | 4.83 | 4.03 | 10.42 | 8.83 | 6.06 | 8.87 | 9.14 |
| Minimum (Pmolec.cm⁻²) | 0.60 | 1.68 | 2.78 | 1.98 | 2.91 | 1.47 | 0.82 | 0.60 | 1.31 | 1.41 | 2.10 | 1.44 |
| Maximum (Pmolec.cm⁻²) | 33.59 | 50.06 | 38.46 | 60.13 | 10.83 | 59.31 | 46.50 | 45.76 | 35.66 | 27.47 | 16.81 | 27.28 |
| 1σ (Pmolec.cm⁻²) | 6.09 | 7.24 | 7.72 | 18.99 | 2.28 | 15.26 | 9.14 | 11.11 | 6.50 | 5.54 | 3.33 | 6.95 |
| Number of days with high quality data | 29 | 17 | 14 | 23 | 9 | 19 | 26 | 29 | 25 | 21 | 10 | 13 |

### 3.2 Impact of wind direction and wind speed on NO2 VCDs in Beijing

The wind influences the concentrations of atmospheric constituents, and thus air quality, in several different ways. The wind transports aerosols and trace gases away from their sources and thus disperses them, leading to lower concentrations. This
includes both vertical mixing due to wind-generated turbulence and advection (horizontal transport). This also implies that at very low wind speed the constituents accumulate which leads to enhanced concentrations (e.g., Feng et al., 2014) and, in particular due to (photochemical) reactions, the formation of haze (e.g., An et al., 2019). The wind direction and its history indicate the transport pathways, determined by large scale weather phenomena (e.g., You et al., 2018; He et al., 2018; Hou et al., 2020; Li et al., 2021; Zhao et al., 2022). When the transport pathways cross source regions, the constituents are
transported to the receptor point. Hence, depending on wind direction, the wind transports polluted or clean air over long distances (e.g., Sundström et al., 2012).

To address the effect of advection on the NO₂ tropospheric VCDs at the Beijing RADI site, a polar map has been created of NO₂ tropospheric VCDs versus wind speed and wind direction (Figure 4). Because of the large diurnal variation of the NO₂ VCDs, only data have been selected at the TROPOMI overpass time at 13:00 BJT. The data in Figure 4 show that, overall,
the tropospheric VCDs are smaller during north-westerly winds, in particular when wind speeds exceed about 4 ms⁻¹. This confirms that wind from these directions transport cleaner air to Beijing. As discussed in Section 2.1, the Yanshan and Taihang mountains are situated to the north and west of Beijing and the area between Beijing and the mountains is mainly agricultural with sparse population and little industry. Thus anthropogenic and industrial emissions are small over this area.



Hence, during low wind speeds from this direction, relatively clean air is transported to Beijing. When wind speeds are

higher, northwesterly winds from the Siberian plains bring clean air, thus greatly improving Beijing's air quality.

During southerly winds, the $NO_2$ VCDs are substantially higher. The high industrial activity and associated traffic and high degree of urbanization in the south of Beijing and the NCP leads to high emissions and thus, during southerly winds, transport of large amounts of $NO_2$ to Beijing. In addition, during low wind speed, the pollutants are accumulated resulting in the observed very high pollution levels at low wind speed, in particular during southerly winds.

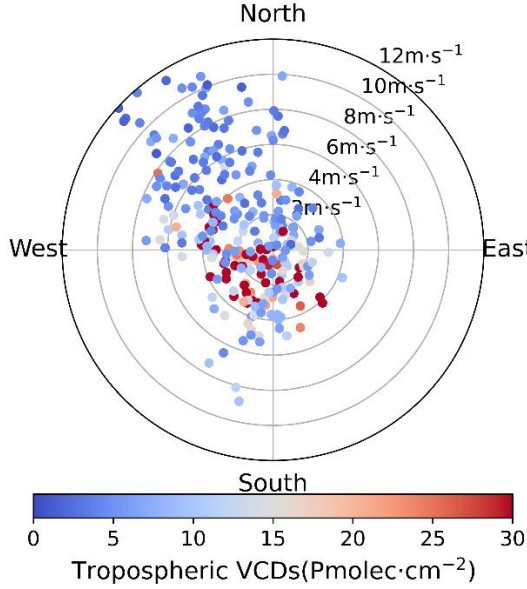


**Figure 4: Polar diagram of $NO_2$ tropospheric VCDs, wind direction and wind speed during the TROPOMI overpass time. Wind speed is indicated by the radius of the circles, ranging from 0 to 12 m.s⁻¹; wind direction corresponds to the angle in polar coordinates, clockwise from north. Note that the south wind direction indicates that the wind comes from the south.**

### 3.3 Tropospheric contribution to total $NO_2$ VCDs

The total $NO_2$ VCDs are determined from the Pandora direct sun measurements and the tropospheric $NO_2$ VCDs are determined from the Pandora sky measurements. Hence the total and tropospheric $NO_2$ VCDs are independently determined, but the tropospheric VCDs should be smaller than or equal to the total VCDs. The data in Figure 2 show the strong diurnal variation of both the total and tropospheric $NO_2$ columns, but the time series of the ratio of the tropospheric to the total $NO_2$ VCD (Figure 5) show that they do not vary in the same way. Because direct sun measurements are made more frequent than

sky measurements, the ratios have been calculated using the first tropospheric VCD sky measurement available after a direct sun measurement and then averaged to obtain daily and monthly means (Figure 6). The results show that the ratio varies between about 0.2 and 1.05, with large diurnal and day-to-day variations, indicating that different processes in the troposphere and above affect $NO_2$ concentrations. The data show that the tropospheric / total ratio decreases to a minimum



around the middle of the day, i.e., early in the day the tropospheric NO$_2$ VCDs decrease faster than the total, before it

increases in the afternoon.

**Figure 5: Time series of the ratios of tropospheric VCDs to total VCDs for each month from August 2021 to July 2022.**


The monthly mean ratios of the $NO_2$ tropospheric and the $NO_2$ total VCDs derived from the Pandora measurements at the Beijing-RADI site are plotted as time series in Figure 6, together with the number of days for which data are available in each month. The results in Figure 6 show that the monthly mean ratios vary between about 0.38 and about 0.75, and there is some inter-monthly variation. Considering only months with 10 or more days for which data are available, the monthly

averages show that the tropospheric $NO_2$ contribution is 50% to 60% in the winter and 60% to 70% in the spring and autumn. The smaller ration in the winter may be related to the frequent occurrence of haze days when tropospheric $NO_2$ is converted to fine particulate matter (e.g., Wang et al, 2020), or to more active photochemical reactions in spring due to enhanced solar shortwave radiation (Zheng et al., 2015; Xie et al., 2015; Cheng et al., 2016). Also, similar to ozone, stratospheric intrusion could be a possible reason for the springtime increase in tropospheric $NO_2$ concentrations (Lin et al., 2015). The standard

deviations in Figure 6 show that ratios are more variable in the spring than in the winter, indicating that day to day variation is larger in the spring, especially in March when the standard deviation was larger than 0.2.

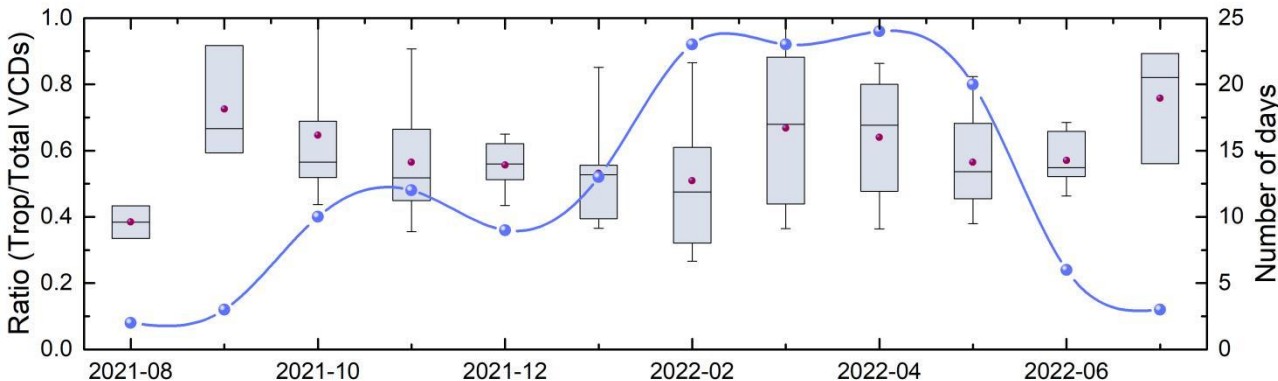

**Figure 6: Box and whisker plot of the monthly mean ratios of $NO_2$ tropospheric VCDs to total VCDs from August 2021 to July 2022, with the number of valid data in each month plotted as a blue dot (right hand axis). For each data point, the dot represents**
**the mean value, the horizontal line represents the median value, the top and bottom edges of the boxes are the 25% and 75% quartiles and the whiskers are the values corresponding to 10% and 90% of the data volume.**

### 3.4 Comparison of Pandora-derived surface $NO_2$ concentrations with ground-based in-situ measurements

During the study period, the Thermo Scientific 42i-TL Analyzer was operated side by side with the Pandora instrument
during a period of 20 days in January 2022. The colocation of these two instruments provides an excellent opportunity for the evaluation of the ground-based $NO_2$ concentrations derived from Pandora sky measurements, using the in situ data from the Thermo Scientific 42i-TL Analyzer as reference. Figure 7 shows the time series of these two data sets. The in-situ observations were made continuously, 24 hours per day, every 10 seconds to 15 minutes, but the Pandora measurements were only made during daylight hours. In addition, there are many gaps in the Pandora time series because only the best





quality data (QA = 0, 1, 10, 11) were retained. To fill part of these gaps, Pandora-derived total VCDs, for which many more

high quality data are available as shown in Figure 2 and discussed in Section 3.1.1, have been included in Figure 7.

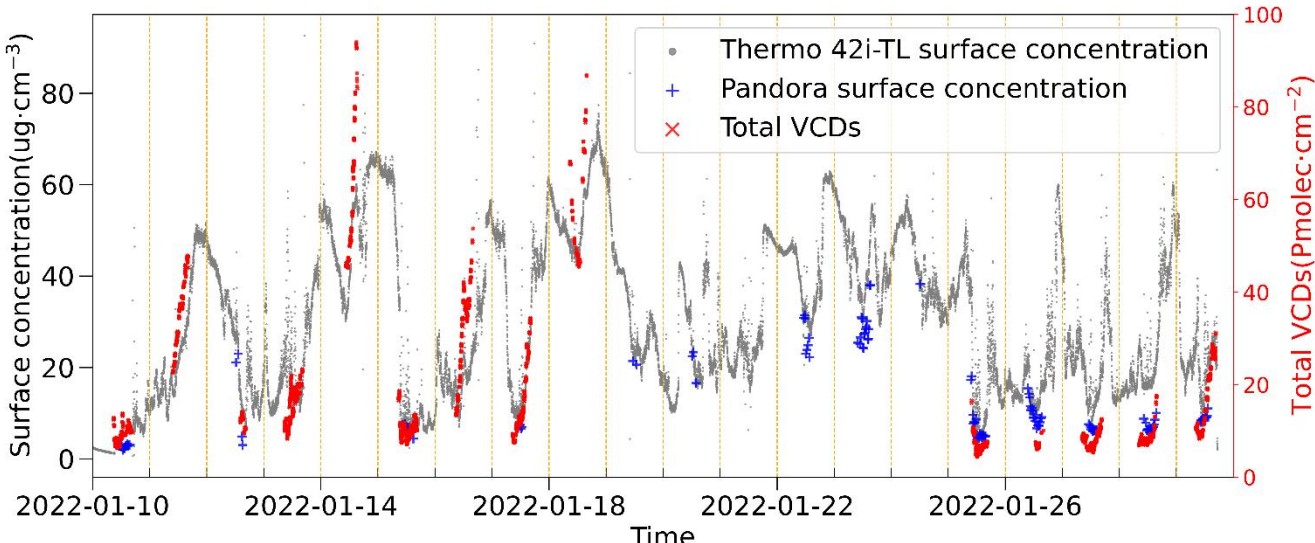

**Figure 7: Time series of NO$_2$ surface concentrations measured using the Thermo Scientific 42i-TL Analyzer (gray) and derived**
**from the Pandora sky data (blue) and total VCDs (red) during 20 days: 10 - 30 January 2022. Yellow vertical dashed lines indicate**
**Beijing Time 0:00. Note that the NO$_2$ total VCDs are plotted on the secondary vertical axis and scaled to match the Thermo 42i-TL**
**Analyzer surface concentration data.**

The continuous time series of the ground-based in situ measurements shows that the NO$_2$ concentrations are usually high

during the night, decrease after sunrise to a minimum in the afternoon and increase thereafter. The observed diurnal variation

of the NO$_2$ concentrations is due to complex photochemical reactions between atmospheric gases such as nitrogen oxides

(NO$_x$), O$_3$, and volatile organic compounds (VOCs) (Law et al., 2002; Xue et al., 2016). On days with large variations of the

NO$_2$ surface concentration, such as on 11, 14, 16 and 17 January 2022 (Figure 7), the total VCDs measured with Pandora

traced the surface concentrations well (note that in Figure 7 the total NO$_2$ VCDs data were scaled to match the Thermo 42i-

TL Analyzer surface concentrations). In addition, the data show that on 11 and 16 January, the tropospheric VCDs increased

earlier than the surface concentrations, suggesting that the increase of the NO$_2$ VCDs on these two days resulted from

atmospheric transport. On 14 January, the surface concentrations changed before the tropospheric VCDs increased,

suggesting the occurrence of an NO$_2$ source near the surface from where NO$_2$ was vertically mixed resulting in the

subsequent increase of the tropospheric VCDs.

The data in Figure 7 further show that the Pandora-derived surface NO$_2$ concentrations are overall relatively low as

compared with the in situ concentrations measured with the Thermo Scientific 42i-TL Analyzer. This is further illustrated in

the scatterplot of the Pandora-derived NO$_2$ surface concentrations versus the Thermo Scientific 42i-TL Analyzer in situ data





(Figure 8). Figure 8 shows that there are two data regimes: for low in situ NO₂ concentrations, up to about 22 µg.cm⁻³, the Pandora-derived surface concentrations increase much slower than the in situ concentrations and thus vary over a much

smaller range (between about 5 and 10 µg.cm⁻³). For larger NO₂ concentrations, the Pandora-derived surface concentrations are closer to the in situ data but with a substantial underestimation. Similar differences have been reported between NO₂ concentrations retrieved from MAX-DOAS measurements, which use a similar technique as used for Pandora to derive surface concentrations, and NO₂ point measurements using instruments mounted in a mast at three heights above the surface (ASL), which, like the Thermo Scientific 42i-TL Analyzer, derive the NO₂ concentrations from chemiluminescence

measurements (Kang et al., 2021). These authors report that MAX-DOAS substantially underestimates the in situ concentrations, by about 65% at 60m and 160 m ASL and by 33% at 280 m ASL, but do not offer an explanation. Below we discuss the Pandora and chemiluminescence data and influencing factors which may contribute to the observed differences.

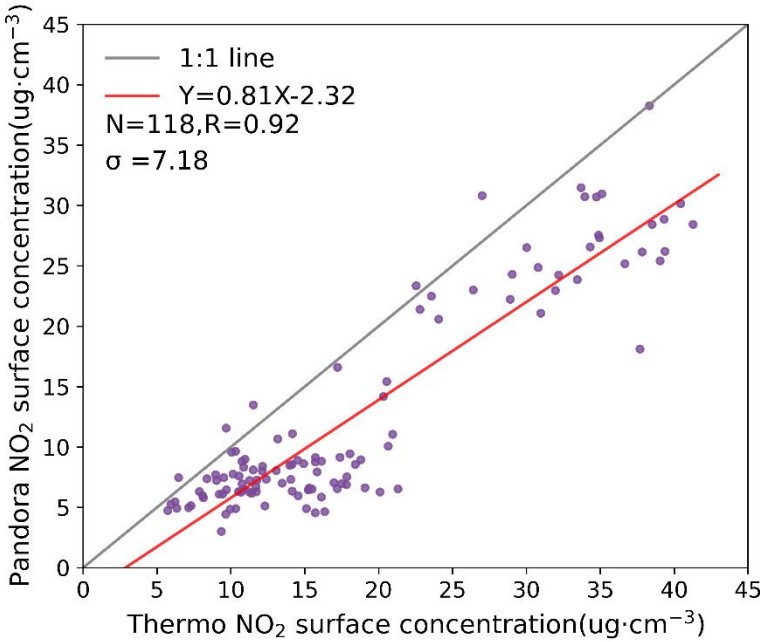

**Figure 8: Scatterplot of NO₂ surface concentrations: Pandora data versus Thermo Scientific 42i-TL Analyzer data.**

The Pandora surface concentration is derived from the ratio of the difference between the SCDs of NO₂ at the largest and smallest SZA and the same difference for air, i.e., $\left(\text{SCD}_{\text{Pandora,max}} - \text{SCD}_{\text{Pandora,0}}\right)/\left(\text{SCD}_{\text{AIR,max}} - \text{SCD}_{\text{AIR,0}}\right)$ (Cede, 2021). In view of the exponential decrease of NO₂ concentrations with height as reported by Kang et al. (2021), the slant optical path

at Pandora's maximum observation angle, which is slightly less than 90°, implies that the NO₂ concentration is integrated over a certain height with a negative NO₂ concentration gradient. As a result, the measured concentration is lower than that at the surface which is measured by the Thermo Scientific 42i-TL Analyzer.





Furthermore, Pandora measures the average $NO_2$ concentration along the (slant) optical path, whereas the Thermo Scientific 42i-TL Analyzer provides a local value at the location of the instrument (Kang et al., 2021) which may be more sensitive to
local variations, such as short-term emissions by passing cars or turbulent perturbations of plumes.

Another factor influencing the comparison of Pandora and in situ measurements of $NO_2$ concentrations using a Thermo Scientific 42i-TL Analyzer, is that this instrument may overestimate atmospheric $NO_2$ concentrations due to interference from other atmospheric constituents which react with ozone to produce chemiluminescence, such as peroxyacetyl nitrate and nitric acid (Dunlea et al., 2007; Steinbacher et al., 2007) or alkenes (Alam et al., 2020). In particular, during the Beijing
winter the nitrate concentrations are high (Luo et al., 2019; Wang et al., 2021) and may contribute to overestimation of the $NO_2$ concentrations measured by the thermo-chemiluminescence method.

### 3.4.1 Difference between pandora and in situ surface $NO_2$ concentrations for different concentration regimes

As mentioned above, Figure 8 shows two groups of data which are separated according to low and high $NO_2$ concentrations. Comparison with the polar diagram in Figure 4 shows that, for the TROPOMI overpass time, the low $NO_2$ concentrations
mainly occur during northerly winds, whereas high concentrations are mainly observed during southerly winds, and more specific for wind directions between about 135º and 315º. The scatterplot in Figure 8 includes all observations for a short time period. To further investigate the different $NO_2$ regimes revealed in Figure 8, wind direction has been added to the time series plot of Figure 7, as well as AOD, and the results are presented in the Supplementary material, Figure S5. Figure S5 confirms that the conclusions from Figure 4, for TROPOMI overpass times, also apply to the observations at other times, i.e.,
that the low surface $NO_2$ concentrations (<10 µg.m$^{-3}$) derived from Pandora almost all occur during northerly winds, are smaller than the in situ concentrations measured at the same time with the Thermo Scientific 42i-TL Analyzer, and trace the latter well. This is well-illustrated on 26 January when the surface concentrations initially decrease (from about 15 and 24 µg.m$^{-3}$ for Pandora and Thermo, respectively) to a minimum, then increase. This variation may be due to a combination of the diurnal variation described in Section 3.1 and the change in wind direction and thus transport pathway. During southerly
winds the Pandora-derived surface $NO_2$ concentrations are substantially higher (>15-20 µg.m$^{-3}$) and increase with increasing in situ data. In particular for in situ concentrations between about 30 and 40 µg.m$^{-3}$ there is a good correlation although Pandora data are still underestimated as discussed above. The data in Figure S5 also show the large changes occurring during changes in wind direction, in combination with diurnal variations.

The difference between the relations between Pandora-derived and in situ measured surface $NO_2$ concentrations for the low
and high $NO_2$ concentration regimes may be explained as follows. During northerly winds, the contribution to the $NO_2$ concentrations from transport over long distances is small because the air masses are transported over relatively clean areas. Thus, the main contribution to the concentration is due to local production of $NO_2$ from anthropogenic activities near the surface, which is first observed by the in-situ sensor. Pandora surface $NO_2$ concentration is an integrated value over the height range covered by the largest SZA. Mixing of the surface-produced $NO_2$ over this height range takes some time, and
thus there is a concentration gradient resulting in a lower value of the Pandora-derived surface concentration. In contrast,





during southerly winds, the air mass transported to the measurement location has passed through an area with high $NO_2$ concentrations and the contribution from local production of $NO_2$ near the surface is relatively small. Thus, the $NO_2$ concentration measured by the in-situ sensor is relatively little enhanced due to local production as compared to that in northerly winds. Therefore, the Pandora-derived surface $NO_2$ concentration compares better to the in-situ data when the
concentrations are high than when they are small.

**3.4.2 Comparison with AOD**

Figure S5 shows that the AOD and $NO_2$ total VCD (both are column integrated quantities measured using collocated ground-based instruments), trace each other very well. Hence the different relationships between MAX-DOAS - and Tower - averaged $NO_2$ concentrations for low and high AOD reported by Kang et al. (2021) are due to the same reason as the
different relationship for low and high $NO_2$ concentrations shown in Figure 8 and discussed above. It is noted that the measurements reported by Kang et al. (2021) were made on a tower located in Beijing at ca. 5 km from the Beijing-RADI site and the conditions at both sites are similar.

**3.5 Validation of TROPOMI $NO_2$ VCDs over Beijing using Pandora**

The Pandora instrument and the Pandora Global Network were designed for satellite validation, i.e., providing VCDs as
independent reference for the validation and evaluation of satellite data. Here Pandora data are used for the validation of TROPOMI $NO_2$ total and tropospheric VCDs, re-sampled to a spatial resolution of $100 \times 100$ $m^2$. Scatterplots of $NO_2$ total and tropospheric VCDs from TROPOMI versus Pandora VCDs and histograms of the TROPOMI-Pandora differences are presented in Figure 9. For TROPOMI, OFFL data were used and for Pandora only high quality data with DQ0, 1, 10 and 11. The Pandora quality control limits the amount of suitable data, especially in June, July, August, and September (Section 3.1).
The scatterplots in Figures 9a and c show the good correlation of both the TROPOMI-derived $NO_2$ total and tropospheric VCD with the Pandora data. For the total VCD, R= 0.95, the slope of the LSQ fit is 0.96 and the RD is -8.44%, well below the expected deviation of $0.5$ $Pmolec \cdot cm^{-2} + (0.2\ to\ 0.5) \cdot VCD_{trop}$. (Van Geffen, 2021). Figure 9a shows that for small values of the total VCDs the data points are very close to the identity line, whereas for VCDs > 10 Pmolec.$cm^{-2}$ the differences are larger and the data points are rather evenly scattered around the identity line. In other words, the uncertainties
in the TROPOMI-derived $NO_2$ total VCD increases with increasing VCD. The histogram of the differences between the TROPOMI and Pandora $NO_2$ total VCDs in Figure 9b shows a uniform distribution with the characteristics of a Gaussian with the mean or median as the symmetry axis. On average, the mean TROPOMI total VCDs are slightly smaller than those from Pandora, with an MD of -1.16 Pmolec.$cm^{-2}$, a MAD of 3.14 Pmolec.$cm^{-2}$ and a standard deviation of 4.44 Pmolec.$cm^{-2}$. This uncertainty is in part due to the large SZA, which leads to uncertainty in the air mass factor (AMF) (Herman et al., 2019;
Griffin et al., 2019; Ialongo et al., 2020). In addition, a 10% error in the AMF estimate due to cloud effects was reported by the ATBD document (Van Geffen, 2021, Van Geffen et al., 2022).

The validation of the TROPOMI-derived $NO_2$ tropospheric VCDs is presented as scatterplots and histograms in Figures 9c and 9d. As for the total VCDs, the TROPOMI tropospheric VCDs trace the Pandora data well and the scatterplot shows the good correlation between both data sets, with a correlation coefficient of 0.92, slightly smaller than for the total VCDs, and a

somewhat smaller slope of 0.94, with a RD of 16.15%. The histogram in Figure 9d shows that the tropospheric VCDs are roughly symmetrically distributed around 0, and the number of large negative deviations (deviation < -10 $Pmolec.cm^{-2}$) is slightly larger than the number of positive deviations (deviation > 10 $Pmolec.cm^{-2}$).

The above analysis confirms the good performance of the TROPOMI $NO_2$ total and tropospheric VCDs and provides confidence for the use of the TROPOMI data. For this analysis, the TROPOMI data were re-sampled to a spatial resolution

of $100 \times 100$ $m^2$, similar to the Pandora observation area.

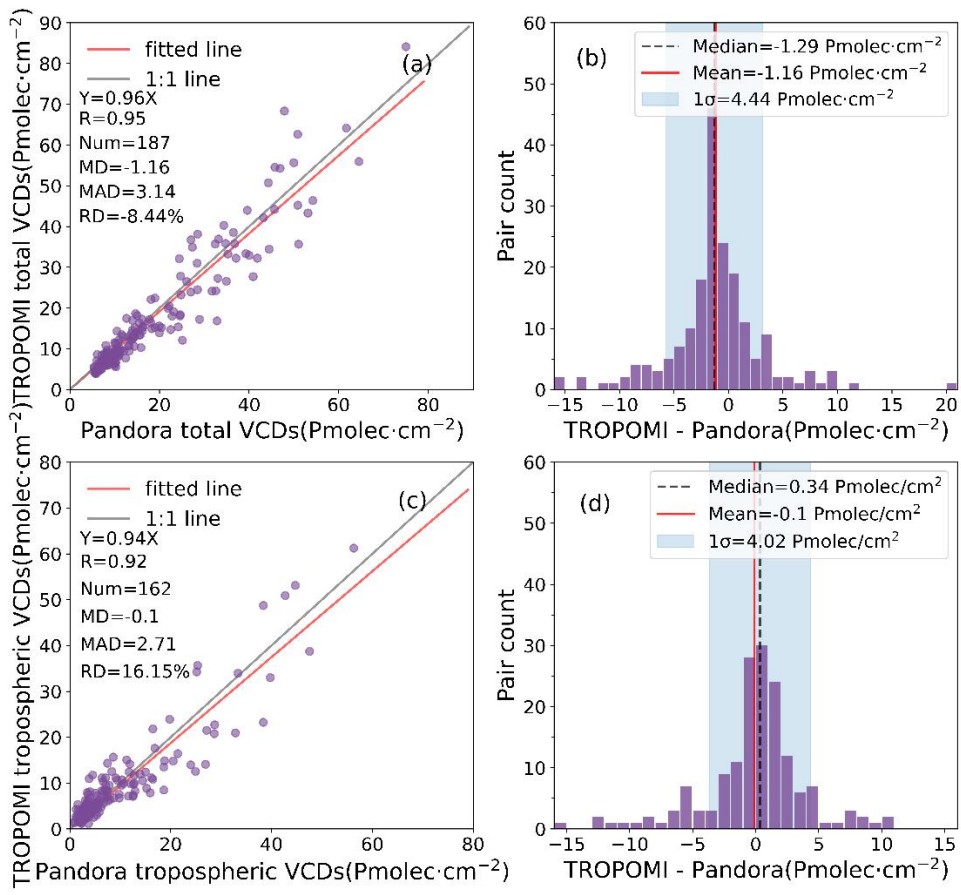

**Figure 9: Validation of TROPOMI total and tropospheric $NO_2$ VCDs, re-sampled to a spatial resolution of $100 \times 100$ $m^2$, using Pandora observation as reference data: (a, c) scatter plots of total and tropospheric TROMPOMI vs Pandora data together with**
**statistical metrics; (b, d) histograms of the differences between TROPOMI and Pandora $NO_2$ total/tropospheric VCDs.**



### 3.6 Spatial representativeness of Pandora vertical column density

Considering the good performance of the TROPOMI-retrieved $NO_2$ VCDs discussed in section 3.5, these data have been used to evaluate the spatial representativeness of the Pandora observations at the Beijing-RADI site. Figure 10a shows the

spatial distribution of the TROPOMI-derived $NO_2$ tropospheric VCD over a wide area centered at the Beijing-RADI site averaged over the first year of Pandora operations (Figure 1). Large differences are observed with high tropospheric $NO_2$ VCDs over Beijing, Tianjin and its highly industrialized surroundings and several other locations. Over the mountains to the north and west of the study area the tropospheric $NO_2$ VCDs are much lower, by a factor of 10 or more. The Beijing-RADI Pandora site is located in the high $NO_2$ tropospheric VCD area (as also shown in the inset of Figure 1), while strong $NO_2$

gradients are observed toward the north and west. Obviously, because of the variation in the $NO_2$ VCD, the correlation between the TROPOMI and Pandora data is influenced by the choice of the area over which the TROPOMI data are averaged. In particular, this may happen when a separation between clean and polluted air occurs over Beijing, as reported, for instance, by Sun et al. (2016) (their Figure 7). The effect of the size of the area over which TROPOMI data are averaged is illustrated in Figure 10b, showing the annual mean (black) and median (red) tropospheric $NO_2$ VCDs averaged over areas

centered at the Beijing RADI Pandora site with increasing radius. The data show that, for this specific case when the TROPOMI $NO_2$ VCDs are averaged over annual mean values, the area-averaged value remains constant within a radius of 10 km from the Pandora site, and decreases when the radius is further increased. This is further illustrated by the data in Table 3, showing that for a radius larger than 10 km, the standard deviation ($\sigma$) increases as the radius increases. For less homogeneous $NO_2$ distributions the area for which the situation at the site is representative will be much smaller than 10 km.

This also implies that the location of the Pandora site influences the validation results. The data in Table 2 show that for a circle of 3 km the $\sigma$ is 0.223 and for a circle of 5 km it is 0.282 Pmolec.cm$^{-2}$. This means that when the Pandora site is at a different location in an overpass pixel, the 3.5×5.5 km$^2$ FOV produces an error in the $NO_2$ tropospheric VCD between 0.223 and 0.282 Pmolec.cm$^{-2}$, i.e., between 1.7% and 2%.





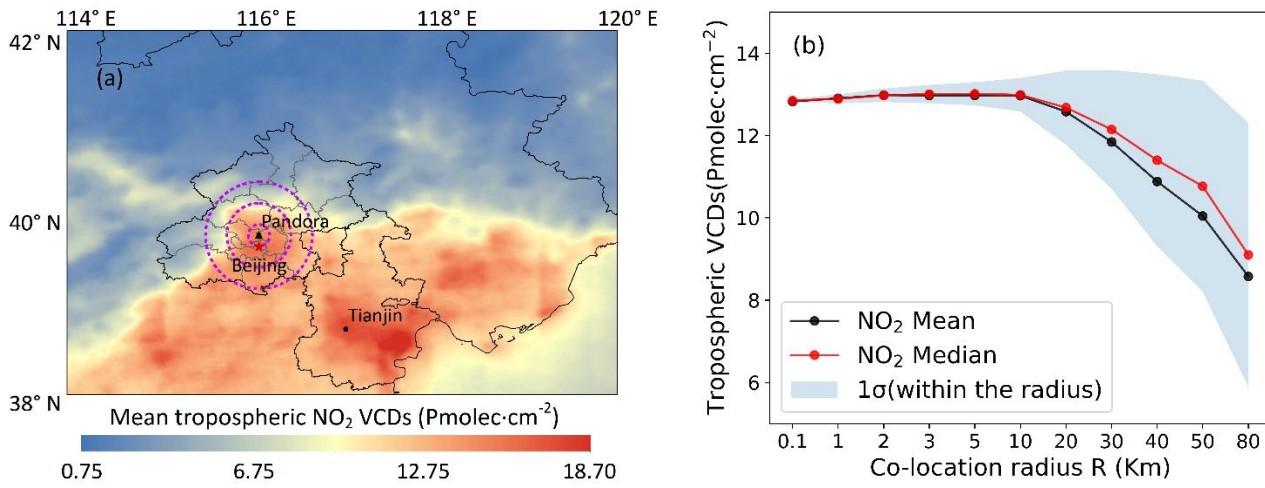

**Figure 10: Illustration of the spatial variation of the NO₂ tropospheric VCD over a wide area around the Beijing-RADI (Pandora-171) site and the evaluation of the representativeness of the Pandora data for a larger area. (a) spatial distribution of the TROPOMI-derived NO₂ tropospheric VCD re-gridded to a resolution of 0.1×0.1 km² over the study area indicated in Figure 1, averaged over the period from August 1, 2021, to July 31, 2022. The three purple circles indicate a radius of 10, 30 and 50 km around the site. (b) annual mean (black) and median (red) tropospheric NO₂ VCD averaged over circles centered at the Beijing RADI Pandora site with increasing radius, as a function of that radius (expressed in km), with variability (1 standard deviation) given as a light blue area around the mean.**

**Table 2: Statistical metrics for the tropospheric VCDs averaged over circles with increasing radius around the Pandora site.**

| Radius (Km) | 0.1 | 1 | 2 | 3 | 5 | 10 | 20 | 30 | 40 | 50 | 80 |
|---|---|---|---|---|---|---|---|---|---|---|---|
| Mean (Pmolec.cm⁻²) | 12.83 | 12.90 | 12.98 | 12.97 | 13.98 | 12.97 | 12.58 | 11.84 | 10.88 | 10.05 | 8.58 |
| Median (Pmolec.cm⁻²) | 12.85 | 12.89 | 12.98 | 13.01 | 13.01 | 12.99 | 12.68 | 12.15 | 11.40 | 10.77 | 9.10 |
| Standard deviation σ (Pmolec.cm⁻²) | 0.037 | 0.110 | 0.164 | 0.223 | 0.282 | 0.407 | 0.901 | 1.444 | 2.087 | 2.563 | 3.210 |
| Dilution factor ($D_f$) | 1 | 1.003 | 1.010 | 1.012 | 1.012 | 1.011 | 0.987 | 0.945 | 0.887 | 0.838 | 0.710 |

## 4 Summary and conclusions

The first operational Pandora site in China has been established on the roof of the laboratory building of the Aerospace Information Research Institute of the Chinese Academy of Sciences (AirCAS) in Beijing, at the end of July 2021. The Pandora instrument provides continuous observations of NO₂ total and tropospheric VCDs together with surface



concentrations from direct sun and sky measurements. The Beijing-RADI site is part of the PANDONIA Global Network (PGN), where the Beijing-RADI data are publicly accessible from the PGN website. In this paper, an overview has been presented of the first year of data from the Beijing-RADI Pandora, i.e., their variations on time scales from hours to seasonal, and the influences of wind speed and direction as well as chemical reactions on these variations. The $NO_2$ surface

concentrations have been compared with independent measurements from a collocated Thermo Scientific 42i-TL Analyzer, using a different physical principle for $NO_2$ concentration measurements. Explanations were offered to explain the differences between Pandora and in situ $NO_2$ surface concentrations. The Pandora $NO_2$ VCD data have been used as an independent reference for the evaluation of TROPOMI retrieved $NO_2$ VCDs over Beijing. This study leads to the following conclusions:


1.  The Pandora observations show that $NO_2$ concentrations in Beijing are high during the winter and low during the summer, with a diurnal cycle where the concentrations reach a minimum during day time; the reduced concentrations during the 2022 Winter Olympics show that the emission control during that period was highly effective.

2.  The contribution of the tropospheric to the total $NO_2$ VCDs varies between 0.2 to 1 with large diurnal to seasonal

variations, is high in the morning and afternoon with a minimum around noon Monthly averages show that the tropospheric $NO_2$ contribution is 50% to 60% in the winter and 60% to 70% in the spring and autumn The relatively small amount of valid data does not allow us to assess the tropospheric contribution in summer.

3.  During northerly winds, the tropospheric $NO_2$ VCD is small, especially when the wind speed is larger than 4 m.s$^{-1}$. When the wind speed is low, clean air from areas between the Yanshan and Taihang mountains is transported to

Beijing. When the wind speed is higher, northwesterly winds transport clean air from the Siberian plains, greatly improving Beijing's air quality. During southerly winds, polluted air is transported from adjacent areas with high $NO_2$ emissions resulting in high $NO_2$ pollution in Beijing. In addition, at low wind speeds, pollutants accumulate which results in increased $NO_2$ concentrations which are most obvious during southerly winds.

4.  Comparison of $NO_2$ surface concentrations derived from Pandora measurements with in-situ data from a Thermo

Scientific 42i-TL Analyzer show that the Pandora-derived concentrations are substantially smaller than the in-situ data. Reasons for this discrepancy have been identified: 1) The Thermo Scientific 42i-TL Analyzer is sensitive to other sources of chemiluminescence contributing to the $NO_2$ signal, such as peroxyacetyl nitrate and nitric acid, or alkenes. 2) The Thermo Scientific 42i-TL Analyzer represents the local in-situ $NO_2$ concentration near the sampling port whereas Pandora represents the average concentration along a slant optical path, instead of a horizontal optical path for near-

surface measurements, because the Pandora observation zenith angle is slightly smaller than 90 degrees. The concentrations are therefore an average over a small vertical range with, considering that the $NO_2$ source is at the surface, a vertical concentration gradient. This results in a an average $NO_2$ concentration which is smaller than the actual concentration at the surface.



5.  The comparison of the Pandora and in-situ surface $NO_2$ concentrations further shows that the data are separated in two clusters with different relations between the Pandora and in situ data: one for low $NO_2$ concentrations during northerly winds where the Pandora concentrations show very little variation with in-situ data and another regime for high $NO_2$ concentrations where the Pandora data are still smaller than the in-situ concentrations but with a definite positive correlation. These differences are explained in terms of transport and local production.

6.  Using the Pandora data for validation of the TROPOMI $NO_2$ total and tropospheric VCD shows the good performance of the TROPOMI retrieval over Beijing with R= 0.95 and 0.92, a LSQ fit with a slope of 0.96 and 0.94 and a standard deviation well below the expected $0.5 \; Pmolec \cdot cm^{-2} + (0.2 \; to \; 0.5) \cdot VCD_{trop}$.

7.  The location of the Pandora instrument within a sub-orbital TROPOMI pixel of $3.5 \times 5.5 \; km^2$ may result in an error in the $NO_2$ tropospheric VCD between 0.223 and 0.282 $Pmolec.cm^{-2}$, i.e., between 1.7% and 2%. The analysis shows that the Pandora observations at the Beijing-RADI site are representative for an area with a radius of 10 km.

**Data availability.**

The TROPOMI data are available via

https://developers.google.com/earth-engine/datasets/catalog/COPERNICUS_S5P_OFFL_L3_NO2.

The Pandonia data are available via http://data.pandonia-global-network.org/.

The ERA5 reanalysis data are available via https://cds.climate.copernicus.eu/.

**Auth contributions.**

OL, YL, CF and GL were involved in the research design. OL, GL, YZ and CF analysed the data. OL, GL, YZ, CF and YL prepared the manuscript. JD provides support to the subsequent data analysis. KL operated and managed the Beijing-RADI Pandonia measurements. PZ, TZ and YW provide data of Thermo Scientific 42i-TL instrument. ZL and YZ provided critical comments which substantially improved the paper. All authors have read and agreed to the published version of the manuscript.

**Competing interests.**

The authors declare that they have no conflict of interest.



**Financial support.**

This work was supported by the National Distinguished Youth Foundation of China (Grant No. 41925019) and National Natural Science Foundation of China (Grant No. 42101365).

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
