# Peer review of "Evaluation of the first year of Pandora NO2 measurements over Beijing and application to satellite validation"

_Atmospheric Measurement Techniques, 2023_

## Author Comment (AC1)

**Reply to referee #1**

The authors thank the referee for the valuable time spent to thoroughly read the manuscript and provide valuable comments which contributed to improvement of this revised version. Below we provide our point-to-point responses, together with the revisions made, where appropriate.

(Referees' comments in red, author responses in black, and adjustments of manuscript in blue.)

The authors report the $NO_2$ observations in Beijing China from first Pandora instrument, show the local temporal variation of $NO_2$ and reveal the spatial and temporal representativeness of atmospheric column $NO_2$ concentrations obtained from ground-based remote sensing. The manuscript is well structured and logic, gives some observational facts and valuable conclusions, and deserves to be published in the AMT journal. However, I still have a few comments as below I hope authors will clarify before publication.

1.  In section 2.2, two subheadings 2.2.3 appear. Also the reanalysis data are not instrumental and should not be presented in this section; the authors are requested to adjust them.

    Thank you for this comment, the second 2.2.3 was revised to 2.2.4, and we changed the section heading: '2.2 Instrumentation' to '2.2 Instrumentation and auxiliary data'.

2.  This manuscript focuses on the analyses and comparisons of pandora instrumental observations and is not solely a measure of the differences between TROPOMI and pandora observations; The paper shows that the pandora observations are also compared to ground-based observations at least and that the differences are measured. It is therefore recommended that the methods section be revised and improved by correcting the description of the paragraph below line 225 and adding the description of the methods in the other sections, if any.

    Thank you for this comment. We fully agree that we use both satellite data and in situ observations in this study. This is also clearly mentioned throughout the manuscript (abstract, introduction, and all other Sections 2, 3 and 4, with subsections to Sections 2 and 3 devoted the parts of the study where satellite and/or in situ data are used). All this is included in "Evaluation" as we used it in

the title and we specifically mention "satellite validation" in the title as an additional aspect because this is an important reason for establishing the Pandora Network. For the evaluation by comparison with independent data sets, like satellite and in situ data, it is common to use statistical metrics and these are summarized in Section 2.3, after a discussion of collocation criteria. Realizing that the correlation coefficient had been mentioned but not defined, we have added a brief description to the text in line 241:
"

The Pearson correlation R (Pearson., 1895) is defined in Eq. (1).

$$R = \frac{\sum_{i=1}^{n}\left(VCD_{TROPOMI,i}-\overline{VCD_{TROPOMI}}\right)\left(VCD_{Pan,i}-\overline{VCD_{Pan}}\right)}{\sqrt{\sum_{i=1}^{n}\left(VCD_{TROPOMI,i}-\overline{VCD_{TROPOMI}}\right)^2}\sqrt{\sum_{i=1}^{n}\left(VCD_{Pan,i}-\overline{VCD_{Pan}}\right)^2}} \quad , \tag{4}$$

"

The description of the collocation and statistical metrics used is applied in several sections and therefore should be in methods, to avoid repeating.
According to the Referee#1's comments, we have added some words at the beginning of Section 2.3 which now reads "For the evaluation of the NO$_2$ observations different methods are used, such as time series to show the variability on different time scales or the effects of external parameters such as wind speed, averaging to reduce short-time variability, scatterplots for comparison with independent data sets. For the comparison between Pandora and TROPOMI NO$_2$ VCDs, the data need to be collocated."

3. In line 252 for the ratio of DQ2 data to total data, 2176 divided by 80,153 does not equal 28.2%. Please check and revise.
Thanks for this comment. We have rechecked the numbers of data in line 261, and 21767 was misspelled as 2176. The associated text in manuscript has been revised to " Among the total VCDs, 21767 data points out of a total of 80153 (27.2%) are low quality. ".

4. In Figure 2 we can see that the number of observations changes from month to month. How is this variation taken into account in the statistical process, e.g., by calculating the median, mean, etc.? Do you average all the observations within a time period in a month or divide the data into per days first and then take the mean?
Thank you for this comment. In our manuscript, indeed we first divided the data into per day and then take the mean to avoid effects on the monthly means due to differences in the number of days when observations were available. This is now also clarified in the text by adding "monthly mean tropospheric VCDs

(calculated from daily averages)" (line 294 and in the caption of Figure 3 we have added "Monthly mean data were calculated from daily means.".

5. In section 3.2, the authors may have missed a phenomenon. There are still several red dots distributed in the north-west around the interval 270° to 320° in Fig 4. However, the author states that clean air is transmitted from the northwest. I think this may not be a coincidence and would appreciate an explanation.

Thank you for this comment. Indeed a small number of red dots, indicating high $NO_2$ concentrations, occur in the North-west wind sector. We have looked at these data points in more detail and came up with the following explanation which has been included in the text (lines 340-347) "As a result, the $NO_2$ concentrations in the northwesterly wind sector are generally low, as shown in Figure 4. However, the data in Figure 4 show some exceptions when $NO_2$ concentrations are high. Further analysis shows that these observations were all made during the winter and are likely due to $NO_x$ emissions from natural gas companies located in the Changping district in the northwest of Beijing. Natural gas is provided for, e.g., heating in the winter, and $NO_x$ is produced during the combustion process (Pan et al., 2023). Thus, in the winter, during northwesterly winds, $NO_2$ is transported to the Pandora site. This explains the observations of high $NO_2$ concentrations, due to local emissions. (More details of high concentration number and time during north-west wind please see Table S2.)"

Table S2: Period of high $NO_2$ concentration during north-westerly winds during winter.

| Time | Tropospheric $NO_2$ VCD (Pmolec.cm$^{-2}$) |
|---|---|
| 2021/11/14 12:32:22 | 32.8 |
| 2021/11/17 13:16:32 | 48.7 |
| 2021/11/24 12:44:22 | 31.7 |
| 2021/12/14 11:27:36 | 41.5 |
| 2021/12/14 13:09:05 | 41.5 |
| 2021/12/19 13:15:25 | 33.1 |
| 2021/12/22 12:18:21 | 29.9 |
| 2021/12/28 12:05:44 | 28.6 |

| 2022/1/1 12:31:14 | 25.1 |
|---|---|
| 2022/3/8 11:58:46 | 29.9 |

6. In section 3.6, why the spatial representation of Pandora is 10km instead of 20km, I noticed that the $D_f$ mentioned in this manuscript is very close between 10km and 20km, with a difference of only 0.002. What is the significance of the author's introduction of $D_f$ if it is not to be used as a metric for evaluation? I would be grateful if this was clarified.

Thank you for your suggestion. Actually, $D_f$ is a reference metric. In our method, we combine both $D_f$ and standard deviation as an indicator of spatial representativeness. The first step is to do an initial screening of spatial representativeness based on $D_f$,: $D_f$ did not change between 1 and 10 km (value 1,011 +/- 0.001) but after 20 km $D_f$ had changed to 0.987, i..e the between 10 and 20 km $D_f$ had changed much more (by 0.024), see Table 2. In Table 2 we also see that at 20 km the standard deviation is twice as large as at 10 km, so we consider this not acceptable. In addition, therefore, in Section 3.6, the spatial representation of Pandora is 10km.

Pan, H., Geng, S., Yang, H., Zhang, G., Bian, H., and Liu, Y.: Influence of H2 blending on NOx production in natural gas combustion: Mechanism comparison and reaction routes, International Journal of Hydrogen Energy, 48, 784-797, https://doi.org/10.1016/j.ijhydene.2022.09.251, 2023.

---

## Author Comment (AC2)

**Reply to referee #2**

The authors thank the referee for the valuable time spent to thoroughly read the manuscript and provide valuable comments which contributed to improvement of this revised version. Below we provide our point-to-point responses, together with the revisions made, where appropriate.

(Referees' comments in red, author responses in black, and adjustments of manuscript in blue.)

This paper presents the $NO_2$ observations from the Pandora spectrometer in Beijing from August 2021 to July 2022. The authors quantitively discuss the temporal variations of $NO_2$ observations on different time scales, and analyze the influences of the wind on $NO_2$ VCDs using reanalysis data. The Pandora $NO_2$ measurements are compared with ground-based in-situ measurements, and the reasons behind their differences are further explained. Finally, the authors use the Pandora $NO_2$ data to validate TROPOMI v1.4 tropospheric $NO_2$ VCDs, and give an estimation of the spatial representativeness of Pandora $NO_2$ measurements.

Overall, I think this paper is clear and well structured. I recommend it be published after addressing the comments listed below.

**Specific Comments:**

1.Line 237-239: Why do you use the defined "standard deviation" instead of covariance, which seems more appropriate, to evaluate TROPOMI and Pandora data sets?

Thank you for this comment. The evaluation metrics are described in Section 2.3 Methodology and are used throughout the manuscript. We have considered your suggestion and also checked some relevant papers (Verhoelst et al, 2020; Ialongo et al., 2021; Lange et al., 2023) that they all use Pearson correlation R. Therefore we decided to continue using the Pearson correlation R and add the equation for R in Section 2.3 (lines 241-242):

"The Pearson correlation R (Pearson., 1895) is defined in Eq. (4).

$$R = \frac{\sum_{i=1}^{n}\left(VCD_{TROPOMI,i}-\overline{VCD_{TROPOMI}}\right)\left(VCD_{Pan,i}-\overline{VCD_{Pan}}\right)}{\sqrt{\sum_{i=1}^{n}\left(VCD_{TROPOMI,i}-\overline{VCD_{TROPOMI}}\right)^2}\sqrt{\sum_{i=1}^{n}\left(VCD_{Pan,i}-\overline{VCD_{Pan}}\right)^2}} \, , \qquad (4)$$

"

2. Line 294: The number of days for December is reduced to 12 or 9? The "Number of days with high quality data" for December in Table 1 is 9.

We thank the reviewer for pointing this out. We rechecked the numbers in our script output as well as the text in the tables and the main text, and we have corrected "the number of days is reduced from 31 to 12" to "the number of days is reduced from 31 to 9" in line 302 of this manuscript.

3. Line 323-324: "Because of the large diurnal variation of the $NO_2$ VCDs, only data have been selected at the TROPOMI overpass time at 13:00 BJT". The causal relationship here is unreasonable. If you select the TROPOMI overpass time, it is expected that you also analyze the TROPOMI $NO_2$ VCDs for comparison. Please consider adding the comparison results with TROPOMI observations, or explaining reasonably your thoughts about the time selection.

Thank you for this comment. Indeed this wording may lead to misunderstanding. The $NO_2$ concentrations (here tropospheric VCD) are very variable and plotting all data throughout a day for analysis of effects of certain parameters influencing the concentrations is not conclusive because the effects are hidden in much larger diurnal variations. Therefore, we selected a certain time to exclude the diurnal effect, and selected 13:00, around the time when the daily concentration is at a minimum. That this is also is the TROPOMI overpass time, but that is not relevant. The sentence has been changed to: "To separate effects of wind direction and wind speed on $NO_2$ VCDs from the large diurnal variations (Figure 2), only data have been plotted at 13:00 BJT when the concentrations are close to their daily minimum.". We also referred to the TROPOMI overpass time in section 3.4 and have removed these words (line 332-334).

4. Line 348-350: what are the reasons for the diurnal variations of the tropospheric / total ratio? Are they related to the diurnal variations of $NO_x$ emissions, photochemistry or stratospheric $NO_2$? Please specify.

Thank you for this comment. The goal of this paper is to describe the first year of observations of $NO_2$ using Pandora and we noticed the difference in time series of tropospheric and total VCDs, which are determined by independent methods. Therefore, we decided to plot the ratio of tropospheric /total and describe the results. Likely these provide information on atmospheric processes,

but we do not know which process is most important. We did a literature search but did not find an explanation for our observations: For line 367-369, "The data show that the tropospheric / total ratio decreases to a minimum around the middle of the day, i.e., early in the day the tropospheric $NO_2$ VCDs decrease faster than the total, before it increases in the afternoon." This is due to a combination of processes, and we added to the text (lines 369-374): "This is due to a combination of processes including sources and sinks, of (photo)chemical nature (Herman et al. 2009), transport influenced by meteorological phenomena such as variations in wind speed and wind direction (discussed above in Section 3.2) and variations in boundary layer height, while also the temperature profile changes throughout the day, influencing reaction rates and chemical balance (Kang et al., 2022). Likely all of these are different between the troposphere and above and therefore influence the ratio tropospheric / total $NO_2$ VCD and its daily evolution."

We also changed the second sentence of Section 3.3 to (lines 358-359) "Hence the total and tropospheric $NO_2$ VCDs are independently determined, and can be used to obtain information on atmospheric processes."

5. Line 362-363: please specify the reasons why enhanced solar shortwave radiation (more active photochemical reactions) can result in higher ratio in the spring.

Thank you for this comment. This sentence was not accurate. What we are trying to describe is that the larger change in standard deviation is due to enhanced solar radiation in the spring, not that the higher absolute values are due to enhanced radiation. We have corrected the original text in the manuscript to (lines 384-391):

"The smaller ratio in the winter may be related to the frequent occurrence of haze days when tropospheric $NO_2$ is converted to fine particulate matter (e.g., Zheng et al., 2015; Xie et al., 2015; Wang et al, 2020), whereas the larger ratio in spring may be derived from reduced stratospheric concentrations due to enhanced solar shortwave radiation (Cheng et al., 2016; Müller, 2021). Similar to ozone, stratospheric intrusion could be a possible reason for the springtime increase in tropospheric $NO_2$ concentrations (Lin et al., 2015), because the higher values in the stratosphere have been observed in many studies (Sioris et al., 2003; Hendrick et al., 2004; Preston et al., 1998). Also, the larger standard deviations in spring (especially in March when it was larger than 0.2) indicate a

larger day-to-day variability than in other seasons, which may be related to more active photochemical reactions in response to enhanced radiation intensity."

6. Line 395-399: How do you know the variations of the tropospheric $NO_2$ VCDs from Figure 7? Only total VCDs are shown in this figure.

Thank you for this comment. We need to apologize for a mistake in the text of this paragraph. We described the observations plotted in Figure 7, i.e. total VCD as we wrote the first time, but by mistake we wrote tropospheric VCD in the following lines. In the revised version this has been corrected. (lines 414-424)

7. Section 3.6: The quantification of the spatial representativeness of the Pandora observations at the Beijing-RADI site is based on TROPOMI v1.4 tropospheric $NO_2$ VCDs. However, it has been well known that TROPOMI v1.x data are significantly underestimated, especially for polluted regions. Please use updated TROPOMI PAL v2.3.1 or reprocessed TROPOMI v2.4.0 tropospheric $NO_2$ VCDs to validate the robustness of your conclusion.

Thank you for this comment. We have checked the TROPOMI version we used. Our study covers the period 1 August 2021 to 31 July 2022. For the period from 1 August 2021 to 14 November 2021, the data were processed using version 2.2.0. For the period from 15 November 2021 to 17 July 2022, the data were processed using version 2.3.1. For the period 18 July 2022 onwards, the data were processed using version 2.4.0. The ATBD (Cede, 2021) mentions that the change from version 1.4 to version 2.2 is a big upgrade due to the treatment of surface albedo. As noted by Referee#2 and also in the TROPOMI $NO_2$ ATBD (van Geffen, et al., 2022b), the underestimation, especially in contaminated areas, has been weakened since version 2.2. Unfortunately, when we did this study, only the versions mentioned above were available during the indicated time periods at the indicated website (see data availability).
We have added the above information on the data versions to the text 'Section 2.2 Instrumentation and auxiliary data' (lines 193-195):
"The TROPOMI OFFL data used in this study were retrieved using retrieval processor version 2.2.0 from 1 August 2021 to 14 November 2022, version 2.3.1 from 15 November 2021 to 17 July 2022 and version 2.4.0 after 18 July 2022 (van Geffen et al., 2022a, van Geffen et al., 2022b)."

**Technical comments:**

8.  Figure 2 and Figure 5: I find these two figures are nearly impossible to read the characteristics of diurnal variations. If the discussion of diurnal variations is important, please add additional figures clearly showing the details.

Thank you for this comment. We agree that Figure 2 and 5 are difficult to read and do not show detail, except for the overall variation. However, they only serve to present an overall overview of the data and the diurnal variation is discussed in Section 3.4. Therefore we have selected 2 months (November and December) as examples which we enlarged to landscape and moved Figure 2 to the Supplementary as Figure S1. This is explained in the text in lines 266-267: "The data in Figure 2 show the total VCDs are larger than the tropospheric VCDs. The data also show the diurnal variation of these parameters and the surface concentrations derived from Pandora.".

The first sentence of Section 3.1.1 has been changed to "The data in Figure 2 show the total VCDs are larger than the tropospheric VCDs. The data also show the diurnal variation of these parameters and the surface concentrations derived from Pandora."

Likewise, Figure 5 has been moved to the Supplementary as number S6 and we only show February as an example in Figure 5. This is explained in the text in lines 360-361: "the time series of the ratio of the tropospheric to the total $NO_2$ VCD (see e.g., Figure 5, where the ratios for February are shown as an example, and Figure S6 for all months)"

9.  Line 35: add a period after "$VCD_{trop}$".

    Thank you for this comment. The text "…error for TROPOMI of $0.5\ \mathrm{Pmolec} \cdot \mathrm{cm}^{-2} + (0.2\ \text{to}\ 0.5) \cdot VCD_{trop}$" have been corrected to "…error for TROPOMI of $0.5\ \mathrm{Pmolec} \cdot \mathrm{cm}^{-2} + (0.2\ \text{to}\ 0.5) \cdot VCD_{trop}$." in line 36.

10. Line 123: change "capitol" to "capital".

    Thank you for this comment. The text in Line 123 have been revised to "…as the capital of China" in line 123.

11. Line 312: change "NO2" to "$NO_2$" in the subheading.

    Thank you for this comment. NO2 has been changed to $NO_2$ in line 321, and we

have checked also the rest of the text.

12. Line 361: change "ration" to "ratio"

Thank you for this comment. The text in line 384 has been corrected to "The smaller ratio in the winter may be related to…".

Ialongo, I., Virta, H., Eskes, H., Hovila, J., and Douros, J.: Comparison of TROPOMI/Sentinel-5 Precursor $NO_2$ observations with ground-based measurements in Helsinki, Atmos. Meas. Tech., 13, 205-218, 10.5194/amt-13-205-2020, 2020.

Kang, H., Zhu, B., de Leeuw, G., Yu, B., van der A, R. J., and Lu, W.: Impact of urban heat island on inorganic aerosol in the lower free troposphere: a case study in Hangzhou, China, Atmos. Chem. Phys., 22, 10623–10634, https://doi.org/10.5194/acp-22-10623-2022, 2022.

Lange, K., Richter, A., Schönhardt, A., Meier, A. C., Bösch, T., Seyler, A., Krause, K., Behrens, L. K., Wittrock, F., Merlaud, A., Tack, F., Fayt, C., Friedrich, M. M., Dimitropoulou, E., Van Roozendael, M., Kumar, V., Donner, S., Dörner, S., Lauster, B., Razi, M., Borger, C., Uhlmannsiek, K., Wagner, T., Ruhtz, T., Eskes, H., Bohn, B., Santana Diaz, D., Abuhassan, N., Schüttemeyer, D., and Burrows, J. P.: Validation of Sentinel-5P TROPOMI tropospheric $NO_2$ products by comparison with $NO_2$ measurements from airborne imaging DOAS, ground-based stationary DOAS, and mobile car DOAS measurements during the S5P-VAL-DE-Ruhr campaign, Atmos. Meas. Tech., 16, 1357–1389, https://doi.org/10.5194/amt-16-1357-2023, 2023.

Müller, R.: The impact of the rise in atmospheric nitrous oxide on stratospheric ozone : This article belongs to Ambio's 50th Anniversary Collection. Theme: Ozone Layer, Ambio, 50, 35-39, 10.1007/s13280-020-01428-3, 2021.

Pearson, K. Notes on Regression and Inheritance in the Case of Two Parents, Proceedings of the Royal Society of London, 58, 240-242. https://doi.org/10.1098/rspl.1895.0041, 1895.

Sillman, S. and He, D.: Some theoretical results concerning O3-NOx-VOC chemistry and NOx-VOC indicators, 107, ACH 26-21-ACH 26-15, https://doi.org/10.1029/2001JD001123, 2002.

Verhoelst, T., Compernolle, S., Pinardi, G., Lambert, J.-C., Eskes, H. J., Eichmann, K.-U., Fjæraa, A. M., Granville, J., Niemeijer, S., Cede, A., Tiefengraber, M., Hendrick, F., Pazmiño, A., Bais, A., Bazureau, A., Boersma, K. F., Bognar, K., Dehn, A., Donner, S., Elokhov, A., Gebetsberger, M., Goutail, F., Grutter de la Mora, M., Gruzdev, A., Gratsea, M., Hansen, G. H., Irie, H., Jepsen, N., Kanaya, Y., Karagkiozidis, D., Kivi, R.,

Kreher, K., Levelt, P. F., Liu, C., Müller, M., Navarro Comas, M., Piters, A. J. M., Pommereau, J.-P., Portafaix, T., Prados-Roman, C., Puentedura, O., Querel, R., Remmers, J., Richter, A., Rimmer, J., Rivera Cárdenas, C., Saavedra de Miguel, L., Sinyakov, V. P., Stremme, W., Strong, K., Van Roozendael, M., Veefkind, J. P., Wagner, T., Wittrock, F., Yela González, M., and Zehner, C.: Ground-based validation of the Copernicus Sentinel-5P TROPOMI $NO_2$ measurements with the NDACC ZSL-DOAS, MAX-DOAS and Pandonia global networks, Atmos. Meas. Tech., 14, 481–510, https://doi.org/10.5194/amt-14-481-2021, 2021.

---

## Author Comment (AC3)

**Reply to referee #3**

The authors thank the referee for the valuable time spent to thoroughly read the manuscript and provide valuable comments which contributed to improvement of this revised version. Below we provide our point-to-point responses, together with the revisions made, where appropriate.

(Referees' comments in red, author responses in black, and adjustments of manuscript in blue.)

Review of Liu et al. "Evaluation of the first year of Pandora $NO_2$ measurements over Beijing and application to satellite validation"

This paper introduces one year of $NO_2$ data measured by the AIRCAS Pandora site in Beijing. The diurnal and seasonal changes are analyzed. Comparisons with in-situ and satellite observations of $NO_2$ are carried out. The impact of atmospheric transport and the emission control policies have also been examined. This paper is well structured and the topic fits with the AMT journal.

Here are my comments:

1. In the abstract, need to brief mention that the Pandora data include total and tropospheric $NO_2$ VCD.

   Thank you for this suggestion. To be more clearly and in accordance with the reviewer concerns, we have added an interpretation in abstract, text "In this paper, an overview is presented of the Pandora $NO_2$ data collected during the first year of operation, i.e., from August, 2021, to July, 2022." has been revised to "In this paper, an overview is presented of the Pandora total and tropospheric $NO_2$ vertical column densities (VCDs) and surface concentrations collected during the first year of operation, i.e., from August, 2021, to July, 2022." in line 21-23.

2. Line 18, full name for "VCD" at its first appearance.

   Thank you for this comment. We have written "vertical column densities (VCDs)" at first appearance in both the abstract (line 22) and the main text (line 85).

3. Line 28-29, how about winter?

Thank you for this comment. We apologize for the mistake in the abstract and corrected the sentence to reflect the results presented in Section 3.3. It now reads "The contribution of tropospheric NO₂ to the total NO₂ VCD varies significantly on daily to seasonal time scales, i.e., monthly averages vary between 50% and 60% in the winter and between 60% and 70% in the spring and autumn." in line 28-30.

4.  Line 35, define the unit "Pmolec"

Thank you for this comment. We have added "$(\,1\,\mathrm{Pmolec}\cdot\mathrm{cm}^{-2} = 1\times 10^{15}\,\mathrm{molec}\cdot\mathrm{cm}^{-2})$." in line 36 of the abstract at the first occurrence. In the main text, first occurrence of $\mathrm{Pmolec}\cdot\mathrm{cm}^{-2}$ was at the end of Section 2.2.1, where it was explained: "$(\,1\,\mathrm{Pmolec}\cdot\mathrm{cm}^{-2} = 1\times 10^{15}\,\mathrm{molec}\cdot\mathrm{cm}^{-2} = 3.745\times 10^{-2}\,\mathrm{DU\ (Dobson\ Unit)} = 7.639\times 10^{-7}\mathrm{kg}\cdot\mathrm{m}^{-2})$ (Herman et al., 2009)." (lines 171-172)

5.  Line 123, capital

Thank you. We changed the text "···as the capitol of China" in line 123 to "···as the capital of China".

6.  Line 171-173, there are two precision estimates mentioned here, are they both for the total VCD product? Also, please explain the DU unit here. Please also quantify the retrieval error in fraction, which is the error divided by the mean NO₂.

Thank you for this comment. In fact, the first one is precision and the second one is accuracy. Considering that readers' understanding of precision and accuracy can easily lead to bias and that accuracy is more concerned about Pandora also reader, we only retain relevant descriptions of accuracy in the manuscript.
As to the "Dobson Unit (DU)", it is usually used unit to indicate how much of a given trace gas, especially $O_3$, is in the atmosphere, and is also commonly used in many research for $NO_2$. The Dobson unit is defined as the thickness (in units of 10 μm) of that layer of pure gas which would be formed by the total column

amount at standard conditions for temperature and pressure. To have a better understanding of DU, we add '(Dobson Unit)' after DU and an International System Unit, kg/m$^2$, in manuscript.

The absolute value of the deviation will change in different scenarios, such as high-concentration and low-concentration scenarios, so here we use the mean fraction of retrieval error in typical cases from Pandora official paper.

Corresponding text in line 171-173 now reads 'The estimated nominal accuracy is about $\pm 2.67 \ \mathrm{Pmolec \cdot cm^{-2}}$ (error in fraction for 5.33%) . ( $1 \ \mathrm{Pmolec \cdot cm^{-2}} = 1 \times 10^{15} \ \mathrm{molec \cdot cm^{-2}} = 3.745 \times 10^{-2} \ \mathrm{DU(Dobson \ Unit)} = 7.639 \times 10^{-7} \mathrm{kg \cdot m^{-2}}$) (Herman et al., 2009).'

7. Section 2.2.1, the total NO$_2$ VCD retrieval from direct sun measurement is straightforward to understand. However, the retrieval of tropospheric partial column using scattered light with different angles is not. Please explain here how the tropospheric component is retrieved from the measurements. What are the assumptions for this retrieval?

In this method for getting the tropospheric VCD of trace gas, such as O$_3$, NO$_2$, HCHO, used by ground-based MAX-DOAS instruments, is assumed that the stratospheric slant column density (SCD) only varies with SZA and is independent of the pointing zenith angle (PZA) of MAX-DOAS. With this assumption, the contribution of stratospheric SCD to total SCD can be removed by subtracting the SCD at PZA=0 from the SCD at PZA=$\alpha$, also referred to as the differential slant column density method. Using the air mass factor (AMF) calculated with an atmospheric model, the SCD can then be converted to the tropospheric VCD. In (Cede et al., 2021) (the Pandora ATBD), the method for removing the contribution from stratospheric, mentioned above, is also employed. Please note, Pandora assumes that O$_2$ concentrations in atmosphere are weakly variable and therefore could be used to characterize atmospheric background concentrations. Differ from MAX-DOAS, Pandora uses both differential slant column density O$_2$ observed by Pandora and climatological O$_2$ VCD as the atmospheric air background to obtain NO$_2$ volume ratios rather than AMF, hence it is considered the variation of atmospheric density, and therefore also atmospheric boundary layer height. To have a better understanding of the retrieval principle of Pandora, we extract the relevant formulas from Pandora algorithm using the schematic in the following figure:

[Figure]

Schematic of Pandora observation geometry.

Cede et al., (2021) use the following formula to obtain the tropospheric vertical column density ($VCD_{trop,NO_2}$).

$$VCD_{trop,NO_2} = \frac{\left(SCD_{75,NO_2} - SCD_{0,NO_2}\right) \cdot VCD_{CLM,AIR}}{SCD_{75,AIR} - SCD_{60,AIR} + VCD_{CLM,AIR}} \qquad (1)$$

Where $VCD_{trop}$ is NO₂ tropospheric VCDs.

$SCD_{75,NO_2}$ is slant column density at PZA 75° observed by Pandora.

$SCD_{75,AIR}$ is the slant column density of air-gas (background air) with PZA=75° observed by Pandora.

$VCD_{CLM,AIR}$ is the total VCDs of air-gas calculated by climatology, and is the integral of effective air-gas height and climatological concentration in per layer of total column.

According to the triangle function theorem, $Cos60° = \frac{VCD_{CLM,AIR}}{SCD_{60,AIR}}$, then:

$$= \frac{\left(SCD_{75,NO_2} - SCD_{0,NO_2}\right) \cdot VCD_{CLM,AIR}}{SCD_{75,AIR} - \dfrac{VCD_{CLM,AIR}}{Cos60°} + VCD_{CLM,AIR}} \qquad (2)$$

$$= \frac{\left(SCD_{75,NO_2} - SCD_{0,NO_2}\right) \cdot VCD_{CLM,AIR}}{SCD_{75,AIR} - VCD_{CLM,AIR}} \qquad (3)$$

$$= \frac{SCD_{75,NO_2} - SCD_{0,NO_2}}{\frac{SCD_{75,AIR}}{VCD_{AIR}} - 1} \tag{4}$$

$$= \frac{SCD_{75,NO_2} - SCD_{0,NO_2}}{\frac{1}{Cos75} - 1} \tag{5}$$

Based on the Equation $VCD_{trop} = \frac{dSCD_{\alpha \neq 0} - dSCD_{\alpha=0}}{AMF_{\alpha \neq 0} - AMF_{\alpha=0}} = \frac{\Delta SCD}{\Delta AMF} = \frac{\Delta SCD}{\frac{1}{\cos\alpha} - \frac{1}{\cos 0°}} = \frac{\Delta SCD}{\frac{1}{\cos\alpha} - 1}$ from (Tian et al., 2019) for MAX-DOAS of retrieval method of tropospheric VCD, hence, the above proof is completed.

8. Section 2.2.3, ERA5 has wind data for different heights, what altitude do you use for you analysis and why?

   Thank you for this comment. Here we followed earlier publications (Ialongo et al., 2020; Zhao et al., 2022) where wind data at four pressure levels (925, 950, 975 and 1000 hPa) were averaged. We added the following text: "ERA5 hourly wind speed data at four pressure levels: 925, 950, 975 and 1000 hPa) were downloaded from the ECMWF website (ERA5 web: https://cds.climate.copernicus.eu/cdsapp#!/dataset/reanalysis-era5-pressure-levels?tab=form, last accessed: 11th July 2023) and averaged (following, e.g., Stein et al., 2015; Ialongo et al., 2020; Zhao et al., 2022) for use in the analysis presented below." (Section 2.2.4, lines 213-217) (as advised by Referee#1 the section numbering has been changed).

9. Section 3.1.1, "The $NO_2$ VCDs decrease in the morning to reach a daily minimum around local noon and then increase." Does this just depend on the diurnal change of boundary layer height, can you see the contribution from traffic/industries? The anthropogenic should play an important role here in regulating the diurnal change.

   Thank you for this comment. The diurnal variation is briefly addressed in Section 3.3 in relation to the variable contribution of tropospheric $NO_2$ to the total $NO_2$ VCD and in Section 3, 4 where we compare Pandora-derived surface concentrations with in situ measurements. In response to your comment we refer to the text in the first paragraph below Figure 7. We further note that other papers report enhanced $NO_2$ concentrations during local rush hours, i.e. from high traffic intensity (Herman et al., 2009; Liu et al., 2023; Di Bernardino et al.,

2023). In our data this signal may be present but is relatively weak in comparison to other studies.

Referee# 2 also commented on the observations in Section 3.3. and in response we added the following text (lines 369-374): " This is due to a combination of processes including sources and sinks, of (photo)chemical nature (Herman et al. 2009), transport influenced by meteorological phenomena such as variations in wind speed and wind direction (discussed above in Section 3.2) and variations in boundary layer height, while also the temperature profile changes throughout the day, influencing reaction rates and chemical balance (cf. Kang et al., 2022, for a brief overview). Likely all of these are different between the troposphere and above and therefore influence the ratio tropospheric / total $NO_2$ VCD and its daily evolution." Disentangling these processes would require a detailed model study, but this is out of the scope of this study.

10. In figure 4, please indicate the height of the wind speed. You may need to show wind speed at different elevation.

Thank you for this comment. In this study we only considered the wind speed and wind direction at the height level for 925, 950, 975 and 1000hPa, referenced from Ialongo et al., 2020; Zhao et al., 2022. This comment is also similar to your comment 8, you also could see our response to that comment there.

11. Line 361, ratio

Thank you for noting this. We have corrected this typo.

12. Line 363-364 "Also, similar to ozone, stratospheric intrusion could be a possible reason for the springtime increase in tropospheric $NO_2$ concentrations (Lin et al., 2015)". This is highly speculative. Can you provide paper reference that show more solid evidences?

Thank you for pointing this out. We note that there have been a number of outstanding studies on $NO_2$ tropospheric and stratospheric profiles. They report that there is a zone of high values of $NO_2$ at 20-30 km (Sioris et al., 2003; Hendrick et al., 2004; Preston et al., 1998), whereas most of the tropospheric $NO_2$ is distributed within the boundary layer, about 1-2 km below (Lin et al., 2014; Wang et al., 2019). Therefore, when the bottom of the stratosphere

collapses with the tropopause in spring (concurrent with the ozone stratospheric intrusion), there may be an effect on the tropospheric column concentration of $NO_2$. Although $NO_2$ is quickly converted to secondary pollutants (ozone, particulate matter, etc.) by photochemical reactions, the conversion of $NO_2$ may be lower than within that in the boundary layer due to the relatively low concentrations of VOCs at the top of the troposphere. Certainly, near-surface anthropogenic emissions result in a dominant $NO_2$ concentration within the boundary layer (~15 ppb, Wang et al., 2019), while stratospheric $NO_2$ concentrations (~5-8 ppb, Preston et al., 1998) may only contribute to tropospheric $NO_2$ concentrations to some extent. However, we would like to state that this is one possible reason for the increase in spring RATIO, and we do not exclude other reasons for this phenomenon, e.g., due to enhanced stratospheric $NO_2$ photolysis by solar radiation in spring. We have modified the original text in the manuscript for greater rigorous as follows in lines 384-391:

"The smaller ratio in the winter may be related to the frequent occurrence of haze days when tropospheric $NO_2$ is converted to fine particulate matter (e.g., Zheng et al., 2015; Xie et al., 2015; Wang et al, 2020), whereas the larger ratio in spring may be derived from reduced stratospheric concentrations due to enhanced solar shortwave radiation (Cheng et al., 2016; Müller, 2021). Similar to ozone, stratospheric intrusion could be a possible reason for the springtime increase in tropospheric $NO_2$ concentrations (Lin et al., 2015), because the higher values in the stratosphere have been observed in many studies (Sioris et al., 2003; Hendrick et al., 2004; Preston et al., 1998). Also, the larger standard deviations in spring (especially in March when it was larger than 0.2) indicate a larger day-to-day variability than in other seasons, which may be due to more active photochemical reactions in response to enhanced radiation intensity."

13. Line 480: although the absolute values increase with increasing $NO_2$ VCD, the fraction (bias divided by the mean $NO_2$) may be similar.

Thank you for pointing this out. The absolute difference is growth with the increasing of $NO_2$ concentration, but, the fraction, a good evaluation metric, may be not similar to absolute difference but changes little/weak. Given that previous text in section 3.5 have descripted the trend of comparison between TROPOMI and Pandora for Figure 9. Hence, we decide to remove the text, 'In

other words, the uncertainties in the TROPOMI-derived NO$_2$ total VCDs increase with increasing VCD.', out of manuscript.

14. Line 554, please mention that "0.2" and "1" are fractions (so they are unitless).

Thanks for your comment. Text "…varies between 0.2 to 1 with large diurnal to …" has been revised to "The fraction of tropospheric NO$_2$ contributing to the total NO$_2$ VCDs varies between 0.2 to 1, with large diurnal to seasonal variations, is high in the morning and afternoon with a minimum around noon." in line 578-579.

Di Bernardino, A., Mevi, G., Iannarelli, A. M., Falasca, S., Cede, A., Tiefengraber, M., and Casadio, S.: Temporal Variation of NO$_2$ and O3 in Rome (Italy) from Pandora and In Situ Measurements, 14, 594, 2023.

Hendrick, F., Barret, B., Van Roozendael, M., Boesch, H., Butz, A., De Mazière, M., Goutail, F., Hermans, C., Lambert, J. C., Pfeilsticker, K., and Pommereau, J. P.: Retrieval of nitrogen dioxide stratospheric profiles from ground-based zenith-sky UV-visible observations: validation of the technique through correlative comparisons, Atmos. Chem. Phys., 4, 2091-2106, 10.5194/acp-4-2091-2004, 2004.

Lin, J. T., Martin, R. V., Boersma, K. F., Sneep, M., Stammes, P., Spurr, R., Wang, P., Van Roozendael, M., Clémer, K., and Irie, H.: Retrieving tropospheric nitrogen dioxide from the Ozone Monitoring Instrument: effects of aerosols, surface reflectance anisotropy, and vertical profile of nitrogen dioxide, Atmos. Chem. Phys., 14, 1441-1461, 10.5194/acp-14-1441-2014, 2014.

Liu, S., Cheng, S., Ma, J., Xu, X., Lv, J., Jin, J., Guo, J., Yu, D., and Dai, X.: MAX-DOAS Measurements of Tropospheric NO$_2$ and HCHO Vertical Profiles at the Longfengshan Regional Background Station in Northeastern China, 23, 3269, 2023.

Sillman, S. and He, D.: Some theoretical results concerning O3-NOx-VOC chemistry and NOx-VOC indicators, 107, ACH 26-21-ACH 26-15, https://doi.org/10.1029/2001JD001123, 2002.

Stein, A. F., Draxler, R. R., Rolph, G. D., Stunder, B. J. B., Cohen, M. D., and Ngan, F.: NOAA's HYSPLIT Atmospheric Transport and Dispersion Modeling System, Bulletin of the American Meteorological Society, 96, 2059-2077, https://doi.org/10.1175/BAMS-D-14-00110.1, 2015.

Sioris, C. E., Haley, C. S., McLinden, C. A., von Savigny, C., McDade, I. C., McConnell, J. C., Evans, W. F. J., Lloyd, N. D., Llewellyn, E. J., Chance, K. V., Kurosu, T. P., Murtagh, D., Frisk, U., Pfeilsticker, K., Bösch, H., Weidner, F., Strong, K., Stegman, J., and Mégie, G.:

Stratospheric profiles of nitrogen dioxide observed by Optical Spectrograph and Infrared Imager System on the Odin satellite, 108, https://doi.org/10.1029/2002JD002672, 2003.

Tian, X., Xie, P., Xu, J., Wang, Y., Li, A., Wu, F., Hu, Z., Liu, C., and Zhang, Q.: Ground-based MAX-DOAS observations of tropospheric formaldehyde VCDs and comparisons with the CAMS model at a rural site near Beijing during APEC 2014, Atmos. Chem. Phys., 19, 3375-3393, 10.5194/acp-19-3375-2019, 2019.

Wang, Y., Dörner, S., Donner, S., Böhnke, S., De Smedt, I., Dickerson, R. R., Dong, Z., He, H., Li, Z., Li, Z., Li, D., Liu, D., Ren, X., Theys, N., Wang, Y., Wang, Y., Wang, Z., Xu, H., Xu, J., and Wagner, T.: Vertical profiles of $NO_2$, $SO_2$, HONO, HCHO, CHOCHO and aerosols derived from MAX-DOAS measurements at a rural site in the central western North China Plain and their relation to emission sources and effects of regional transport, Atmos. Chem. Phys., 19, 5417-5449, 10.5194/acp-19-5417-2019, 2019.

---

## Author Response (AR2)

Dear Editor,

On behalf of all co-authors, I thank the Editor and the Reviewers for their comments and suggestions which substantially improved our manuscript. In this final version we have followed the suggestion for a minor correction (now at line 195) and added the following in the Introduction at line 104: "The Beijing-RADI Pandora is included in the data suite which is routinely used for TROPOMI S5P validation." We also made some minor editorial changes in lines 266-267.

Sincerely yours,

Ouyang Liu